# Geometry-Aligned Tangent-Plane Diffusion Transformers for 360° Panorama Generation

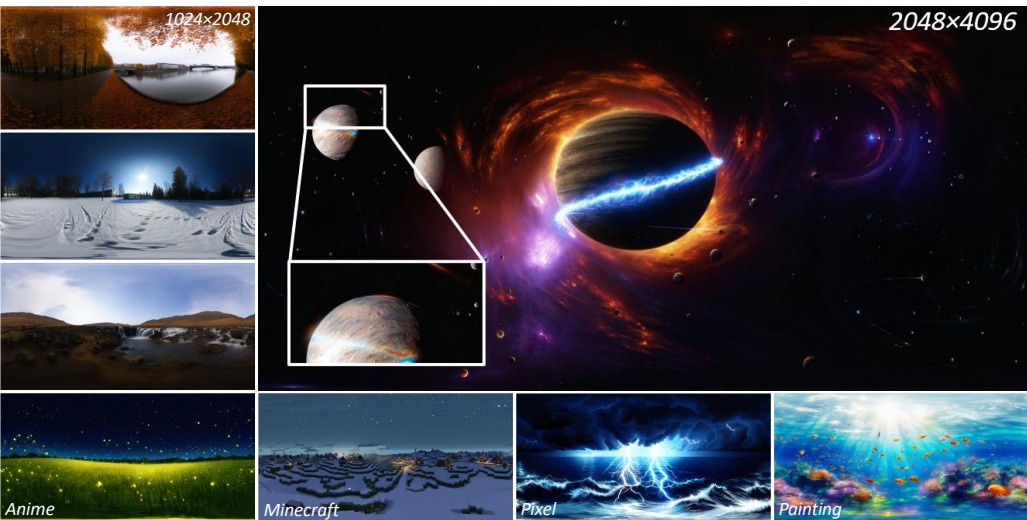

Figure 1: **Panoramic samples generated by TanDiT across resolutions and styles.** TanDiT produces seamless 360° panoramas up to 4K resolution and adapts to diverse artistic styles. By factorizing the sphere into tangent-plane views, it leverages pretrained diffusion transformers and off-the-shelf super-resolution to achieve high-quality, flexible generation without custom architectures.

## Abstract

Generating 360° panoramas from text is challenging due to the inherent difficulty of mapping a 2D diffusion process to a spherical representation without introducing visual artifacts, inconsistencies, or a lack of global coherence. We present TanDiT, a tangent-plane diffusion transformer that factorizes the sphere into locally planar patches, providing a geometry-aligned representation where a pretrained DiT backbone operates without architectural changes. A lightweight ERP-conditioned refinement stage harmonizes overlaps and improves global coherence. To better evaluate panorama quality, we introduce TangentFID and TangentIS, distortion-aware metrics that capture pole and seam degradations, and align closely with human preference. Experiments across multiple benchmarks show that TanDiT outperforms prior work in both perceptual quality and distortion-sensitive fidelity, while scaling efficiently to 4K resolution. Ablations confirm that the main gains arise from the representational choice, establishing TanDiT as a simple and principled framework for high-fidelity panorama generation.

## 1 Introduction

Generating 360° panoramic images from text is central to immersive media, virtual reality, and 3D scene understanding. Unlike perspective synthesis, panoramas must capture the entire spherical field of view, where distortions and wrap-around consistency pose unique challenges. Most existing approaches rely on **equirectangular (ERP)** or **cubemap** projections. ERP-based methods such as Diffusion360 (Feng et al., 2023) and StitchDiffusion (Wang et al., 2023a) reduce seam artifacts

through circular blending or stitch-region denoising, but distort geometry near the poles and misalign receptive fields. Cubemap models (e.g., CubeDiff (Kalischek et al., 2025) reduce distortion but introduce face discontinuities.**Autoregressive strategies** like PanoLlama (Zhou et al., 2025) and PAR Wang et al. (2025) improve wrap-around consistency but suffer from slow inference. Others introduce **spherical operators**, such as SMGD (Sun et al., 2025), or unified omni-directional frameworks like Omni² (Yang et al., 2025), though at the cost of architectural complexity. Despite these advances, seams and poles remain failure points for high-fidelity panoramic generation.

We address this gap by revisiting the **representation**. We introduce **TanDiT**, *a tangent-plane diffusion transformer* that factorizes the sphere into locally planar perspective patches, aligning the geometry of panoramic imagery with the inductive biases of diffusion transformers (DiTs). This geometry-aligned design allows a standard DiT backbone (Peebles & Xie, 2023) to model panoramic structure without architectural modifications. An ERP-conditioned refinement step further harmonizes overlaps and preserves global coherence. A crucial challenge remains, however: the accurate evaluation of panoramic quality. Standard metrics such as FID, or panorama-specific metrics like OmniFID fail to penalize local degradations in high-distortion areas. We propose **TangentFID** and **TangentIS**, distortion-aware metrics that explicitly account for poles and seams, which show strong correlation with human preferences.

**Contributions.** (i) We present **TanDiT**, the first framework that combines tangent-plane factorization with a pretrained DiT, enabling geometry-aligned panorama generation in a single diffusion loop. (ii) We design a **consistency-aware refinement** step that removes seams, enforces loop consistency with circular padding, and stabilizes high-resolution ERPs with negligible overhead. (iii) We propose **distortion-aware metrics**, TangentFID and TangentIS, which use patch-level confidence bounds to capture worst-region failures (poles, seams). We validate them through human studies, showing stronger correlation with perception than ERP- or cubemap-based metrics. (iv) TanDiT achieves superior performance over all baselines on standard and distortion-aware metrics.

## 2 RELATED WORK

**Image Generation.** Diffusion models have become the dominant paradigm for conditional image generation. Models like Stable Diffusion (Rombach et al., 2022) and DALL-E 2 (Ramesh et al., 2022) use U-Nets to predict noise during a denoising process in latent space. More recent work, including Stable Diffusion 3 (SD3) (Esser et al., 2024) and Flux (Labs, 2024), replaces U-Net with transformer-based architectures, offering improved scalability and better modeling of long-range dependencies. These advances motivate our choice of a DiT backbone for panoramic generation.

**Projection-Based Panoramas.** Many methods rely on ERP or cubemap projections. StitchDiffusion (Wang et al., 2023a), inspired by the MultiDiffusion framework (Bar-Tal et al., 2023), introduces overlapping ERP regions and explicit stitching to enforce left–right consistency. PanFusion (Zhang et al., 2024) combines perspective views with an ERP canvas, while Diffusion360 (Feng et al., 2023) and PanoDiff (Wang et al., 2023b) employ circular blending to maintain wrap-around consistency. CubeDiff (Kalischek et al., 2025) generates cubemap faces sequentially via outpainting. UniPano (Ni et al., 2025b) shows that fine-tuning value/output projections in attention layers is effective for panoramic adaptation. These methods improve seams but remain vulnerable to projection distortions, particularly at poles.

**Training-Free, Multi-View Methods.** Several works adapt training-free paradigms. PanoFree (Liu et al., 2024) builds panoramas via iterative warping and inpainting of multi-view images. SphereDiff (Park et al., 2025) extends MultiDiffusion for training-free panorama synthesis, but needs up to 89 tangent-plane views. These approaches reduce retraining cost but scale poorly in computation.

**Geometry-Aware Operators.** Some models directly encode geometry. CurvedDiffusion (Voynov et al., 2023) conditions on lens parameters to simulate distortions, while SMGD (Sun et al., 2025) introduces spherical manifold convolutions, enabling diffusion directly on the sphere. These designs explicitly encode geometric priors but add architectural complexity and custom operators.

**Autoregressive and Decoupled Models.** Autoregressive strategies treat panoramas as sequences. PanoLLaMA (Zhou et al., 2025) sequentially predicts crops at multiple resolutions, while PAR

Table 1: **Comparison of recent 360° panorama generation methods.** Models are grouped by representation and compared in terms of whether they address seam consistency, training requirements, resolution flexibility, and out-of-domain (OOD) generalization. TanDiT uniquely combines tangent-plane representation, DiT backbones, and consistency-aware refinement in a single diffusion loop.

| Model | Representation | Generations | Seam Consistency | Training-Free | Resolution | OOD |
|---|---|---|---|---|---|---|
| PanFusion (Zhang et al., 2024) | ERP | 1 | ✔ | ✗ | Fixed | ✗ |
| StitchDiffusion (Wang et al., 2023a) | ERP | 1 | ✔ | ✗ | Arbitrary | ✗ |
| Diffusion360 (Feng et al., 2023) | ERP | 1 | ✔ | ✗ | Arbitrary | ✗ |
| PanoDiff (Wang et al., 2023b) | ERP | 1 | ✔ | ✗ | Arbitrary | ✗ |
| CubeDiff (Kalischek et al., 2025) | Cubemap | 5 | ✗ | ✗ | Fixed | ✔ |
| SphereDiff (Park et al., 2025) | Tangent Plane | 89 | ✗ | ✔ | Arbitrary | ✔ |
| PAR (Wang et al., 2025) | ERP | Autoregressive | ✔ | ✗ | Fixed | ✗ |
| UniPano (Ni et al., 2025a) | ERP | 1 | ✔ | ✗ | Fixed | ✔ |
| TanDiT (Ours) | Tangent Plane | 1/2 | ✔ | ✗ | Arbitrary | ✔ |

(Wang et al., 2025) adopts masked autoregressive generation with circular padding and alignment. PanoDecouple (Zheng et al., 2025) separates distortion guidance from content completion. These approaches improve consistency but often trade off efficiency.

**Unified Omni-Directional Frameworks.** Omni2 (Yang et al., 2025) proposes a unified framework that handles diverse omni-directional tasks, from synthesis to editing and inpainting. While broader in scope, its effectiveness depends on the availability of large-scale task-specific data, and it does not directly address distortion artifacts inherent to ERP or cubemap training.

**Evaluation metrics.** Standard metrics such as FID and IS are commonly used for generative evaluation but are poorly suited to panoramic images, as ERP-based averaging underweights distortions at poles and seams. OmniFID (Christensen et al., 2024) adapts FID by projecting panoramas into cubemap faces, but this still dilutes localized artifacts and introduces distortions at face boundaries. Our work extends this line by proposing TangentFID and TangentIS, which evaluate in tangent-plane space and explicitly capture worst-case regional failures (see §4.4).

## 3 PRELIMINARIES

**Spherical panoramas.** A panoramic image can be modeled as a function $I : S^2 \to \mathbb{R}^3$ on the unit sphere. The *equirectangular projection* (ERP) maps $(\theta, \phi)$ to a 2D grid but suffers severe polar distortions. The *cubemap projection* reduces distortion by unfolding the sphere into six faces, but introduces discontinuities along face boundaries. Both misalign local neighborhoods, hindering convolutional and transformer-based generative models.

**Tangent-plane projection. Tangent-plane projection.** An alternative is to represent a panorama as a set of tangent-plane patches. Given a reference direction $q \in S^2$, a gnomonic projection maps a spherical direction $s \in S^2$ to coordinates $(u, v)$ on the tangent plane $T_q$:

$$u = \frac{f \, s \cdot t^x}{s \cdot q}, \qquad v = \frac{f \, s \cdot t^y}{s \cdot q},$$

where $(t^x, t^y)$ form an orthogonal basis of $T_q$ and $f$ is focal length. For small field-of-view, tangent patches approximate perspective images with minimal distortion. We denote the tangent-plane decomposition of a panorama as $\mathcal{G} = \{X_k\}_{k=1}^K$.

**Diffusion Models.** Diffusion models (Sohl-Dickstein et al., 2015; Ho et al., 2020) synthesize data by gradually transforming Gaussian noise into a sample through a learned denoising process. The training objective is to predict the injected noise at each step. Modern diffusion models often follow the latent diffusion paradigm (Rombach et al., 2022), which applies the process in a lower-dimensional latent space obtained via a variational autoencoder (VAE), improving efficiency. An alternative is flow matching (Lipman et al., 2023; Liu et al., 2023), which casts training as learning a vector field

for an ODE that continuously maps noise into data:

$$\mathcal{L}_{\text{FM}} = \mathbb{E}_{t,p_t(\mathbf{z}|\epsilon),p(\epsilon)}||v_\theta(\mathbf{z},t) - u_t(\mathbf{z},\epsilon)||_2^2, \quad \mathbf{z}_t = (1-t)\mathbf{x}_0 + t\epsilon \tag{1}$$

These formulations underlie recent models like SD3 (Esser et al., 2024) and Flux (Labs, 2024).

**Diffusion Transformers.** While early diffusion models like DDPM (Ho et al., 2020) and Stable Diffusion (SD) (Rombach et al., 2022) used U-Nets (Ronneberger et al., 2015), recent works leverage the advantages of Diffusion Transformers (DiTs) (Peebles & Xie, 2023). DiTs process image and text features through transformer blocks, using either a double-stream design with cross-attention, or single-stream blocks that jointly process all tokens. SD 3 uses only double-stream layers, while Flux combines both paradigms. DiTs have been shown to model spatial correlations more effectively than U-Nets (Song et al., 2025), a property that we exploit in our geometry-aligned design.

## 4 METHOD

### 4.1 PROBLEM SETUP

Given a text prompt $y$, our goal is to generate a spherical panorama $I : S^2 \to \mathbb{R}^3$. ERP and cubemap projections distort geometry and misalign receptive fields, leading to seam artifacts. We instead adopt a tangent-plane factorization, introduced in prior panorama works (Park et al., 2025), and show that, when combined with a DiT, it yields a principled and efficient alternative.

### 4.2 TANGENT-PLANE FACTORIZATION

**Representation.** Given a text prompt $y$, our goal is to generate a spherical panorama $I : S^2 \to \mathbb{R}^3$. ERP and cubemap projections distort geometry and misalign receptive fields, leading to seam artifacts. We instead adopt a tangent-plane factorization introduced in (Park et al., 2025), and show that, when combined with a DiT, it yields a principled and efficient alternative.

**Grid layout.** Patches are arranged into a 2D grid $\mathcal{G}$, where a permutation $\rho$ places patch $k$ at grid cell $\rho(k)$. To preserve spatial coherence, $\rho$ is optimized to approximate spherical adjacency (see §4.4). Neighboring patches are assigned soft masks $M_k$ that down-weight boundaries, ensuring smooth blending during fusion.

**Training objective.** TanDiT is trained with the *conditional flow matching* (CFM) loss. Let $\mathbf{x}_0$ denote the clean VAE latents of the tangent grid $\mathcal{G}$. At time $t \in [0,1]$, we interpolate between the data and noise distributions as

$$\mathbf{z}_t = (1-t)\mathbf{x}_0 + t\epsilon, \quad \epsilon \sim \mathcal{N}(0, I). \tag{2}$$

The training objective is

$$\mathcal{L}_{\text{CFM}} = \mathbb{E}_{t,\mathbf{z}_t,\epsilon}\left\|v_\theta(\mathbf{z}_t,t,y) - u_t(\mathbf{z}_t,\epsilon)\right\|_2^2, \tag{3}$$

where $v_\theta$ is the DiT-predicted velocity field conditioned on the text embedding $y$, and $u_t$ is the target velocity. This loss matches the conditional flow matching formulation used in recent large-scale DiTs, but is applied directly to tangent-grid latents rather than ERP or cubemap representations.

**DiT backbone.** We build on the pretrained SD3 model (Esser et al., 2024), which replaces the U-Net architecture of earlier diffusion models with a DiT. SD3 offers two advantages that are especially relevant for panoramic synthesis: (i) the transformer backbone captures long-range dependencies more effectively than U-Nets, which is critical for enforcing global coherence across tangent patches, and (ii) large-scale pretraining provides strong generalization, enabling high-quality outputs even when fine-tuned on limited panoramic data. In TanDiT, all DiT weights are fine-tuned end-to-end on tangent grids using the CFM loss. Unlike prior tangent-plane approaches such as SphereDiff (Park et al., 2025), which rely on generating dozens of tangent views in a training-free pipeline, TanDiT leverages a pretrained DiT to produce a full panorama in a single diffusion loop over $\mathcal{G}$. The overall training pipeline is summarized in Fig. 2.

*By combining tangent-plane factorization with a pretrained DiT (SD3), our model achieves geometry-aligned panorama generation in a single diffusion loop, benefiting from both the representational advantages of tangent patches and the long-range modeling capacity of large-scale DiTs.*

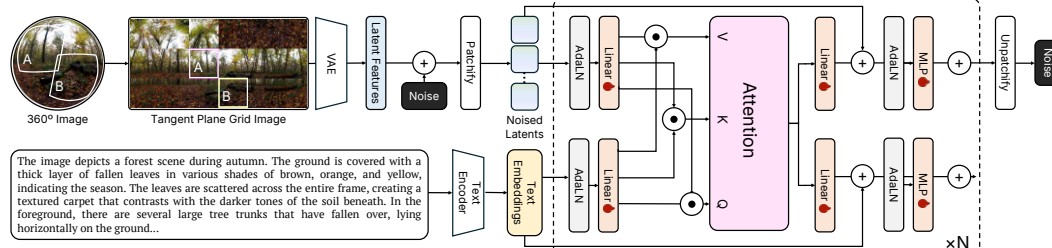

Figure 2: **TanDiT training pipeline.** A 360° panorama is decomposed into perspective-consistent tangent patches and arranged into a grid $\mathcal{G}$ using gnomonic projection. The grid is encoded into latent space and fine-tuned end-to-end with a pretrained DiT under the conditional flow matching objective. Unlike ERP or cubemap inputs, the tangent grid eliminates projection distortions while enabling the DiT to capture global structure and generate full panoramas in a single diffusion loop.

## 4.3 CONSISTENCY-AWARE REFINEMENT

Reprojecting the tangent grid $\mathcal{G}$ into an ERP panorama can expose seams and mismatches in overlapping regions, since the diffusion objective does not explicitly enforce patch-to-patch coherence. To address this, we introduce a lightweight ERP-conditioned refinement stage that harmonizes overlaps while preserving the global layout.

**ERP-conditioned denoising.** We first fuse the tangent patches into an ERP panorama $\hat{I}_{\text{ERP}}$, then encode it into VAE latents $\mathbf{z}_{\text{ERP}}$. To wash out local inconsistencies, $\mathbf{z}_{\text{ERP}}$ is perturbed with Gaussian noise at a high timestep $T_{\text{high}}$ and denoised back to $\mathbf{z}_{\text{refined}}$ using the same DiT backbone conditioned on the text prompt $y$. This step removes high-frequency seams while maintaining scene structure.

**Loop consistency via circular padding.** Direct denoising of ERP latents can introduce discontinuities at the left and right panorama boundaries. Following (Feng et al., 2023; Wang et al., 2023b), we apply circular padding in the horizontal dimension so that edge regions receive meaningful context at each denoising step, ensuring loop-consistent panoramas.

**High-resolution refinement.** When refining panoramas at resolutions beyond 1024×2048, DiTs often degrade near the image boundaries due to aspect ratio imbalance. To mitigate this, we partition the ERP latent into 1024×1024 patches, denoise them independently, and stitch them back together. This strategy makes efficient use of the model's receptive field while enabling high-quality generation at 2048×4096 resolution.

**Inference pipeline.** An overview of the inference process is shown in Fig. 3. Starting from Gaussian noise and prompt $y$, TanDiT generates a tangent grid, reprojects it into $\hat{I}_{\text{ERP}}$, and applies ERP-conditioned refinement with circular padding and super-resolution to produce the final panorama.

*ERP-conditioned refinement harmonizes patch overlaps, enforces loop consistency through circular padding, and stabilizes high-resolution generation, all with negligible additional cost.*

## 4.4 GRID OPTIMIZATION

The arrangement of tangent patches within the grid $\mathcal{G}$ strongly influences how effectively the DiT can model panoramic structure. A naive ordering ignores the geometry of the sphere, often placing adjacent spherical views far apart in the grid. This weakens attention locality and leads to higher reliance on refinement. Further details are found in the supplementary.

**Optimization objective.** We cast grid arrangement as a discrete optimization problem. Let $\rho$ be the permutation assigning each spherical patch $X_k$ to a grid cell. The cost of a permutation is defined as

$$\mathcal{C}(\rho) = \lambda_1 \sum_{(i,j)\in\mathcal{N}} \|\rho(i) - \rho(j)\|_1 + \lambda_2 \sum_{(p,q)\in\mathcal{O}} \|\pi_{\text{ERP}}(p) - \pi_{\text{ERP}}(q)\|_2, \qquad (4)$$

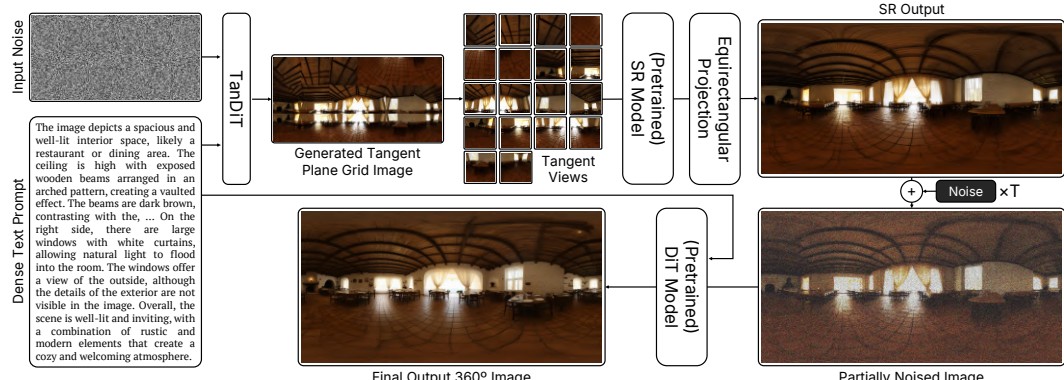

Figure 3: **TanDiT inference pipeline.** From noise and a text prompt, the model generates a tangent grid $\mathcal{G}$, which is fused into an intermediate ERP panorama. An optional super-resolution stage increases resolution, after which ERP latents are perturbed and refined with the same DiT backbone. Circular padding enforces left–right loop consistency, while ERP-conditioned denoising harmonizes overlaps and stabilizes high-resolution outputs, producing seamless 360° panoramas.

where $\mathcal{N}$ is the set of neighboring patches on the sphere (view adjacency term) and $\mathcal{O}$ is the set of pixel pairs that overlap in ERP space (ERP alignment term). The weights $\lambda_1, \lambda_2$ balance adjacency preservation against ERP consistency.

**Optimization algorithm.** We apply simulated annealing to search for low-cost permutations. The algorithm iteratively swaps patch positions in the grid, accepting changes that reduce $\mathcal{C}(\rho)$ and occasionally accepting worse swaps to escape local minima. In practice, this converges quickly to a stable grid ordering.

*The optimized grid ensures that patches close on the sphere remain spatially close in $\mathcal{G}$, while minimizing ERP fusion errors. This improves attention flow in the DiT, reduces boundary inconsistencies, and yields higher fidelity near poles and seams.*

### 4.5 DISTORTION-AWARE METRICS

Global metrics such as FID and IS average over the entire ERP panorama, which systematically underweights pole and seam regions where artifacts are most perceptible. To address this, we introduce **TangentIS** and **TangentFID**, two region-aware metrics operating directly on tangent-plane views.

Given $K$ tangent views $\{X_k\}$, we compute IS and FID independently on each view, yielding distributions of patch-level scores. Rather than averaging, which would again dilute localized errors, we summarize these distributions with confidence bounds:

$$\text{TangentIS} = \mu_{\text{IS}} - 1.96 \cdot \frac{\sigma_{\text{IS}}}{\sqrt{K}}, \quad \text{TangentFID} = \mu_{\text{FID}} + 1.96 \cdot \frac{\sigma_{\text{FID}}}{\sqrt{K}}, \quad (5)$$

where $\mu$ and $\sigma$ denote the mean and standard deviation across tangent views, respectively. The lower bound for TangentIS penalizes poor regional diversity, while the upper bound for TangentFID highlights worst-case fidelity failures. These ensure that good generation requires consistency across all views, not just in the equatorial regions. In §5 and the Appendix, we validate TangentFID and TangentIS, showing high correlation with perceptual judgments, unlike ERP-based FID or OmniFID.

## 5 EXPERIMENTAL SETUP

**Datasets.** We train and evaluate TanDiT on a combination of three datasets: Matterport3D (Chang et al., 2017) (∼10K panoramas), Polyhaven (polyhaven.com, 2025) (∼750 panoramas), and Flickr360 (Cao et al., 2023) (∼3K panoramas). None of these datasets provide text captions, so we generate them automatically using LLaVA-OneVision (Li et al., 2025) to produce dense scene descriptions. For models with limited text context, we further provide concise summaries generated

Table 2: **Quantitative comparison of panoramic image generation models across standard and proposed metrics.** TanDiT demonstrates consistently strong performance across all metrics, reflecting its ability to generate high-quality, coherent, and well-aligned panoramic images. The best result is highlighted in **bold**, while the second-best result is indicated with an underline.

| Model | FID↓ | KID↓ | IS↑ | CS↑ | FAED↓ | OmniFID↓ | DS↓ | TangentFID↓ | TangentIS↑ |
|---|---|---|---|---|---|---|---|---|---|
| MultiDiffusion | 79.35 | 0.050 | **7.27** | **26.92** | 4.00 | 88.68 | 0.0050 | 60.16 | 5.48 |
| StitchDiffusion | 69.68 | 0.040 | 4.90 | 22.09 | 8.42 | 108.47 | 0.0021 | 62.74 | 5.29 |
| Diffusion360 | 42.56 | 0.024 | 3.59 | 21.25 | 2.80 | 48.32 | 0.0007 | 37.48 | 5.72 |
| Panfusion | 35.25 | **0.012** | 4.65 | 23.05 | 2.93 | 48.36 | 0.0007 | 39.57 | 5.68 |
| UniPano | 36.52 | 0.016 | 4.64 | 20.38 | 2.65 | 61.89 | 0.0006 | 47.88 | 5.36 |
| TanDiT (Ours) | **31.60** | 0.013 | 4.39 | 24.93 | **1.76** | **47.93** | **0.0003** | **32.83** | **5.88** |

with LLaMA-2 (Touvron et al., 2023), which are also used for CLIP score evaluation (Radford et al., 2021). We construct 18 tangent-plane views per panorama to allow near-lossless reconstruction, and resize each tangent view to 192×192 pixels, forming a 3×6 tangent grid of size 576×1152 pixels.

**Baselines.** We compare TanDiT against recent panorama generation models with public implementations: (1) **ERP-based:** PanFusion (Zhang et al., 2024), StitchDiffusion (Wang et al., 2023a), Diffusion360 (Feng et al., 2023), UniPano (Ni et al., 2025b); (2) **Training-free/multi-view:** MultiDiffusion (Bar-Tal et al., 2023) (not a true 360° model, included for completeness);

**Implementation details.** TanDiT builds on the pretrained SD3 DiT backbone, which we fine-tune end-to-end on tangent grids. Training uses AdamW (Loshchilov & Hutter, 2019) with a fixed learning rate of $10^{-4}$, batch size 8, and bfloat16 precision. Models are trained for 10 epochs on a single NVIDIA A40 GPU (∼40 GPU-hours). For inference, panoramas are refined with our ERP-conditioned denoising stage (§4.3) and upsampled with a pretrained super-resolution model before final decoding. For fairness, we retrain all of the tested baselines on our dataset using their public implementations and recommended settings, ensuring that performance differences are attributable to model design rather than data mismatch.

# 6 RESULTS

## 6.1 QUANTITATIVE COMPARISON

We evaluate models across both conventional and panoramic-specific metrics: FID and KID measure distributional alignment; Inception Score (IS) captures image quality and diversity; CLIPScore (CS) assesses semantic alignment with the text prompt; FAED leverages a panorama-trained autoencoder for feature similarity; OmniFID adapts FID to cubemap projections; and Discontinuity Score (DS) detects seams in ERP images. Our distortion-aware TangentFID and TangentIS explicitly quantify regional failures at poles and seams. Full metric definitions are provided in the supplementary.

Table 2 shows that TanDiT consistently achieves the best or second-best results across all panoramic-specific metrics. Gains are quite pronounced on TangentFID and TangentIS, which highlight improvements in distortion-prone regions. Compared to ERP-based models (PanFusion, StitchDiffusion, Diffusion360, UniPano), TanDiT produces sharper details with fewer polar artifacts. Models which produce images with a lot of detail in the equatorial regions perform well on the perspective metrics (FID, IS, CS). Overall, TanDiT sets a new state of the art in panoramic image generation under both standard and distortion-aware evaluation.

## 6.2 QUALITATIVE RESULTS

Fig. 4 and 5 illustrate TanDiT's qualitative advantages. ERP-based models such as StitchDiffusion and Diffusion360 produce complete panoramas but often show distortions near the poles and visible seams at the left–right boundary. TanDiT, by generating tangent-plane views and applying ERP-conditioned refinement, yields globally coherent panoramas with consistent texture, lighting, and

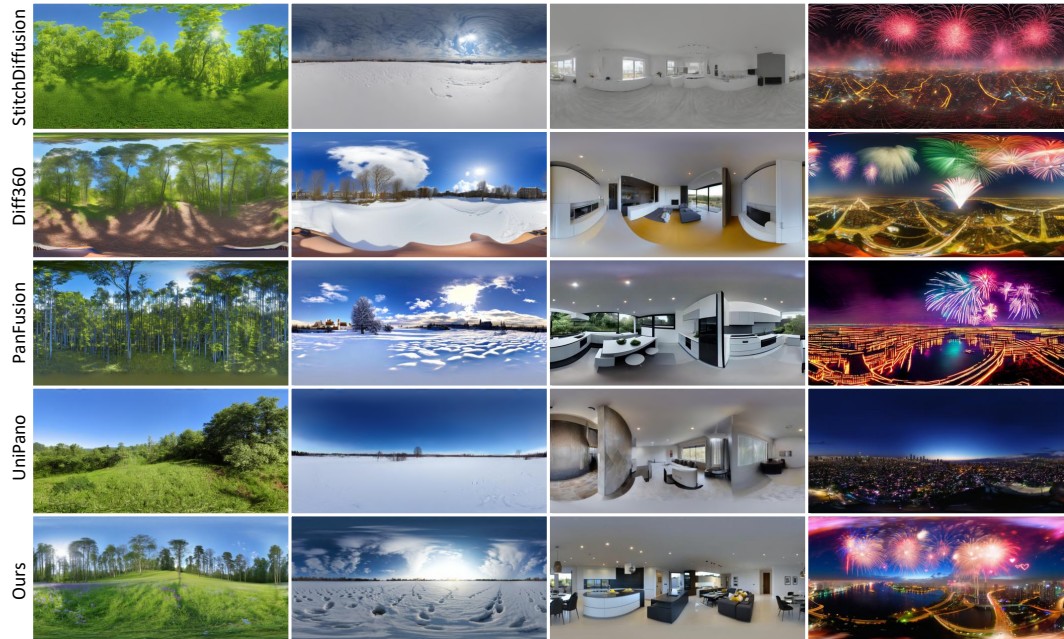

Figure 4: **Qualitative comparisons.** ERP-based methods (StitchDiffusion, Diffusion360) exhibit polar distortions and seam artifacts, while PanFusion and UniPano loses fine detail. MultiDiffusion produces only perspective panoramas. TanDiT generates globally coherent panoramas with sharper details and robust pole/seam consistency, even on challenging out-of-domain prompts (e.g., fireworks).

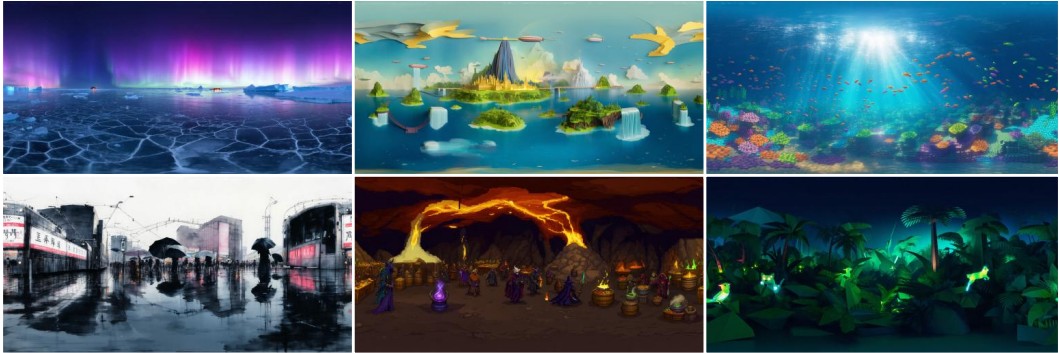

Figure 5: **Style generalization with TanDiT.** TanDiT adapts to diverse stylistic prompts (cinematic, papercraft, minecraft, watercolor and ink, pixel art, low polygon) while preserving panoramic fidelity and spatial coherence.

structure across the full field of view. The fireworks example in Fig. 4 highlights TanDiT's robustness in an out-of-domain setting: while other models lose fine details and continuity, our approach preserves structure and realism. Fig. 5 demonstrates style generalization, where TanDiT adapts to diverse domains, including including cinematic, papercraft, minecraft, watercolor and ink, pixel art, low polygon, while maintaining panoramic fidelity. These results confirm that TanDiT not only improves seam and pole consistency but also generalizes well across prompts and rendering styles.

## 6.3 USER STUDY AND PERCEPTUAL VALIDATION

To assess perceptual quality, we conducted a two-alternative forced choice (2AFC) user study with 30 participants. Each participant was shown 33 pairs of generated panoramas, rendered as 360° rotating videos, and asked to choose the more realistic image based on consistency, detail, and over-

all fidelity. TanDiT was preferred in the majority of comparisons: 86% vs. Diffusion360, 73.3% vs. MultiDiffusion, 78.9% vs. StitchDiffusion, and 78.6% vs. PanFusion, all significant at $p<0.05$ (Wilson interval). These results confirm that users consistently favor TanDiT across diverse prompts and scene types. We also measured the correlation between user preferences and automated metrics. TangentFID and TangentIS showed the highest alignment with human judgments, correctly predicting the model which humans preferred in two different user studies we ran, while ERP-based metrics such as FID and IS often misranked models that failed at poles or seams. This validates our claim that distortion-aware metrics better capture perceptual quality in panoramic generation. We provide a more detailed analysis validating our metrics in the supplementary material.

## 6.4 ABLATION STUDIES

We ablate key components of TanDiT to measure their impact on performance (Table 3). **Circular padding** prevents discontinuities at the ERP left–right boundary. Removing it increases seam artifacts and doubles DS. **Latent rotation** encourages the DiT to learn global coherence rather than to overfit to fixed grid layouts. Without it, TangentIS drops, reflecting weaker diversity across views. **Super-resolution** enables generation at higher resolutions. Removing it lowers texture fidelity and increases FAED and OmniFID. **Patch denoising** stabilizes refinement at high aspect ratios by processing ERP patches independently. Removing it severely degrades performance, with TangentFID nearly doubling. Overall, each module contributes to preserving global consistency, sharpness, and robustness at poles and seams, with patch denoising and circular padding being the most critical.

Table 3: **Ablation study evaluating the impact of key components of TanDiT.** We report the performance of our model's variants with specific components removed. Each ablation leads to a noticeable performance drop, confirming the importance of patched denoising, circular padding, latent rotation, and super-resolution in maintaining global consistency, image quality, and panoramic fidelity. The best result is highlighted in **bold**, while the second-best is indicated with an underline.

| Model | FID↓ | KID↓ | IS↑ | CS↑ | FAED↓ | OmniFID↓ | DS↓ | TangentFID↓ | TangentIS↑ |
|---|---|---|---|---|---|---|---|---|---|
| TanDiT | 31.6 | 0.013 | 4.39 | 24.93 | **1.76** | 47.93 | **0.0003** | **32.83** | **5.88** |
| w/o Circ. Pad. | **31.26** | **0.012** | 4.39 | 24.48 | 1.81 | **47.17** | 0.0013 | 34.07 | 5.36 |
| w/o Lat. Rot. | 31.43 | 0.013 | 4.34 | 24.69 | 1.80 | 47.30 | 0.0005 | 33.66 | 5.49 |
| w/o SR | 35.65 | 0.017 | **4.77** | 24.98 | 2.19 | 57.30 | 0.0003 | 36.61 | 5.71 |
| w/o Patch Denoising | 44.22 | 0.028 | 4.52 | 24.72 | 1.94 | 101.17 | 0.0007 | 87.81 | 4.90 |

## 6.5 4K GENERATION RESULTS

TanDiT generates panoramas as tangent-plane views, thus existing super-resolution models can be applied independently to each view. In our experiments, we apply a $4\times$ scale factor on the default $192\times192$ tangent patches, yielding panoramas up to 4K resolution 2048×4096). Fig. 1 shows an example. The modular design allows scaling to higher resolutions without retraining the backbone, preserving structural consistency while improving detail. This demonstrates TanDiT's practicality for real-world panoramic applications such as VR and 360° content creation.

## 7 CONCLUSION

We introduced TanDiT, a geometry-aligned framework that integrates tangent-plane factorization with pretrained diffusion transformers for high-quality 360° panorama generation. A lightweight ERP-conditioned refinement stage improves seams, poles, and scalability, while distortion-aware metrics (TangentFID, TangentIS) provide perceptually aligned evaluation. TanDiT outperforms all ERP-, and tangent-based baselines across both standard and distortion-aware metrics. We will release code, metrics, and datasets to facilitate fair benchmarking and future research.

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

- Runtime analysis with respect to generation time and resolution scalability
- Memory requirement analysis with respect to the peak GPU memory usage during inference
- Details regarding our optimization algorithm for the optimized grid arrangement
- Additional ablation studies examining the impact of refinement noise levels, grid ordering, tangent plane count, usage of a smaller backbone model, and usage of shorter captions
- Additional qualitative results, including visual comparisons with CubeDiff and visualizations of loop consistency
- Implementation details of our user study and survey design for subjective evaluation
- Discussion of known limitations and ethical considerations of our approach
- License and release plan covering our dataset annotations, benchmark suite, and model checkpoints
- Additionally, we provide a website, accessible through the index.html file, to allow for interactive viewing of many different panoramic image samples.

## APPENDIX

## A  METHODOLOGICAL AND ARCHITECTURAL DETAILS

### A.1  GNOMONIC PROJECTION FORMULATION

The gnomonic projection is a nonconformal mapping (a mapping which doesn't preserve angles) from the sphere onto the plane. We adopt the gnomonic projection to map points from the surface of a sphere to 2D image coordinates $(x, y)$ on a plane tangent to the sphere at a reference point $S$. While the projection is nonconformal, it does preserve straight lines, making it well-suited for generating perspective-consistent tangent plane views.

More specifically, let $S$ be the point of intersection between the sphere and the projection plane. Let $\lambda_0, \phi_0$ be the central longitude and central latitude of the projection (the longitude and latitude of the point $S$). Then, for an arbitrary point with longitude and latitude $\lambda, \phi$, respectively, the resulting coordinates on the plane are given by

$$x = \frac{\cos(\phi) \sin(\lambda - \lambda_0)}{\cos(c)}, \tag{6}$$

$$y = \frac{\cos(\phi_0) \sin(\phi) - \sin(\phi_0) \cos(\phi) \cos(\lambda - \lambda_0)}{\cos(c)}, \tag{7}$$

where

$$\cos(c) = \sin(\phi_0) \sin(\phi) + \cos(\phi_0) \cos(\phi) \cos(\lambda - \lambda_0). \tag{8}$$

Here, $c$ represents the angular distance from the point $(x, y)$ to the center of the projection. Note that the gnomonic projection would map antipodal points to the same location on the sphere, so at minimum, a different plane is required for each hemisphere. The projection can be restricted even further by ensuring that any point with angular distance from the central longitude and central latitude more than half the desired FOV of each tangent plane is not projected.

To achieve full panoramic coverage while maintaining low distortion, we extract a total of 18 tangent views arranged in a fixed $3 \times 6$ grid. The views are evenly distributed across the sphere to balance equatorial and polar regions. Overlapping fields of view between adjacent tangent planes are intentionally included to ensure smooth stitching and promote spatial continuity during training and refinement.

## A.2 EQUIRECTANGULAR-CONDITIONED REFINEMENT

Projecting the generated tangent-plane grid into a panoramic image via equirectangular projection often introduces visible artifacts and inconsistencies, particularly in the overlapping regions between adjacent views. These artifacts stem from limitations of the flow-matching loss used in the DiT backbone, which does not explicitly account for spatial coherence in the reprojected panorama.

To address this, we propose a refinement stage, termed *Equirectangular-Conditioned Refinement*, which leverages the strong generative capacity of the pretrained DiT to improve both the visual consistency and fine details of the panorama. Importantly, prior work has shown that diffusion models are not inherently equipped to handle the geometric distortions present in equirectangular projections. Therefore, we treat the intermediate ERP image as a conditioning signal.

To mitigate the aforementioned issue, we use the previously generated intermediate equirectangular image as a conditioning input to the pretrained DiT during the refinement step. Specifically, we perturb the latent representation of the equirectangular image using noise sampled from a high timestep of the noise scheduler. This perturbation helps suppress high-frequency inconsistencies such as misalignments or localized artifact while preserving the low-frequency structure, including the global scene layout. The model is then conditioned on both this noisy latent and the original text prompt, which guides the denoising process to restore semantic alignment and visual coherence.

However, this refinement step introduces a new challenge: loop inconsistency at the horizontal boundaries of the ERP image, due to a lack of information exchange between the left and right edges. To resolve this, we implement a circular padding strategy (Feng et al., 2023; Wang et al., 2023b), in which the left and right edges of the latent are padded using pixel values from the opposite edges. Specifically, we pad each side with 5 columns from the opposite edge, perform denoising, and then remove the padded regions. This operation is repeated at every denoising step to maintain continuity at the image boundaries. Finally, before decoding the latent into an image, we apply circular padding once more, decode the latent, and crop the padded regions to produce a loop-consistent panorama.

## A.3 CUSTOM EVALUATION METRICS

In this section, we formally justify the design of our proposed TangentIS and TangentFID metrics, and explain their advantages over standard metrics such as Inception Score (IS), Fréchet Inception Distance (FID), and OmniFID when applied to panoramic images.

### A.3.1 FEATURE-BASED JUSTIFICATION

The standard Inception Score suffers from two key issues when used on 360° images. First, the Inception network accepts $299 \times 299$ inputs while equirectangular projections have a 2:1 aspect ratio. Resizing ERP images to fit the required input introduces significant distortion. Second, the Inception model is trained on perspective images, making it inherently biased toward outputs that resemble perspective-style framing such as those produced by wide-angle or NFoV-based models rather than full panoramas.

To further emphasize this, we use the Grad-CAM Selvaraju et al. (2016) approach to identify which regions the Inception features are focused on in the ERP images. A few examples are found in Figure 6a, but we ran the test on 50 such images for proper verification. It is clearly seen that the Inception features focus entirely on the equatorial regions of the image, a bias introduced by the training on perspective images. This means that metrics based on the Inception model aren't properly able to consider polar distortions, or distortions near the edges of the ERP image.

Additionally, we further test the Inception model on spherical rotation. Since an ERP image is left-right consistent, we can apply an arbitrary rotation in the horizontal direction without introducing any seams or discontinuities in the image. The desired result after such a rotation would be that the

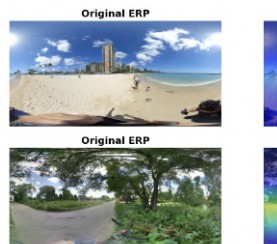 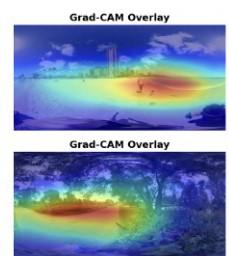

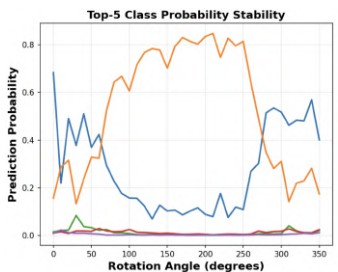

(a) A visualization of several ERP images and their corresponding Inception feature visualization, produced by Grad-CAM. We can clearly see that the Inception model focuses entirely on the center (equator) of the ERP images, and completely neglects the polar regions. Thus, the features will ignore any polar distortions (or lack of polar distortions), and even some distortions on parts of the vertical edges.

(b) Tracking the top 5 probabilities predicted by the Inception model for a given ERP image as we progressively apply higher amounts of horizontal rotation. We see that the rotation causes the features to change, but also causes the predicted class to change completely. This indicates that the Inception features, and thus the IS and FID metrics, can be heavily manipulated by simply applying horizontal rotations.

features stay the same, or at the very minimum, the most likely class stays the same. However, as expected, the Inception model's features change drastically as we increase the amount of rotation, and we see a shift between multiple classes, one example of which can be seen in Figure 6b.

To address this, TangentIS applies the Inception Score independently to each of the 18 tangent-plane images, each sized $192 \times 192$, which are undistorted and perspective-consistent. This preserves spatial fidelity while maintaining compatibility with the Inception model.

Furthermore, as discussed in the main paper, averaging the 18 IS scores corresponding to each tangent plane can unfairly reward models that perform well only in the equatorial regions. For example, wide perspective methods may generate high-quality outputs at the equator but perform poorly near the poles. A simple mean would mask these failures. To address this, TangentIS instead uses the lower bound of the 95% confidence interval across the 18 scores. This penalizes inconsistent models and better reflects the holistic quality of $360°$ generation.

The same limitations apply to FID, as it also relies on the Inception model. However, we further compare our TangentFID to OmniFID, which computes FID across cubemap projections.

### A.3.2 THEORETICAL JUSTIFICATION

The cubemap representation introduces geometric distortion, especially near the edges of each cube face. This distortion increases with distance from the cube center, degrading the reliability of the extracted features. There are 3 relevant types of distortion for all representations: length distortion (the length of a line segment on the sphere vs its length on the projected image, $D_L(\theta)$), angular distortion (the change in angles from the sphere to the projected image, $D_\omega(\theta)$), and area distortion (area on the sphere vs projected area, $D_A(\theta)$). No perspective projection of a sphere can minimize all 3 of these distortions at once. A thorough exploration of different spherical distortions can be found in (Snyder, 1987).

Now, lets compare these distortions on our 18 tangent plane representation vs a cubemap. Table 4 shows a comparison of the different types of maximum distortion for each of the representations. We see that cubemaps always produce a higher distortion compared to the tangent plane representation, but the maximum area distortion is significantly worse for cubemaps ($2.34\times$ higher). In particular, this means that if a cubemap face is generated as a regular perspective image without distortion, and then projected onto the sphere, there will be significant distortions near the edges of each cube face, compared to the same for our tangent plane representation. Conversely, if you want to take an existing spherical image and compute metrics with respect to that image, using a cubemap representation (like OmniFID does) introduces significantly more distortion, which can affect the accuracy of the metrics.

Additionally, the input resolution constraint of the Inception model further limits the effectiveness of cubemap-based evaluation. With cube faces sized at $299 \times 299$, the maximum ERP resolution

Table 4: **A comparison of the different types of distortion introduced by our tangent plane representation compared to the cubemap representation.** Here, $D_L(\theta)$ is the length distortion at angle $\theta$ away from the center (the two numbers represent the distortions of a radial and tangential line, respectively), $D_\omega(\theta)$ is the angular distortion at angle $\theta$ away from the center, and $D_A(\theta)$ is the total area distortion at angle $\theta$ from the center (if we have a cube on the sphere, we assume that one set of sides is radial and the other set is tangential). We can see that the cubemap has worse distortions of every type, but the area distortion is especially bad, over twice the level of distortion as our tangent plane representation.

| Representation | Max $D_L(\theta)$ | Max $D_\omega(\theta)$ | Max $D_A(\theta)$ |
|---|---|---|---|
| 18 Tangent Planes | 1.7/1.3 | 15.24 | 2.22 |
| Cubemap | 3/1.73 | 31.08 | 5.20 |

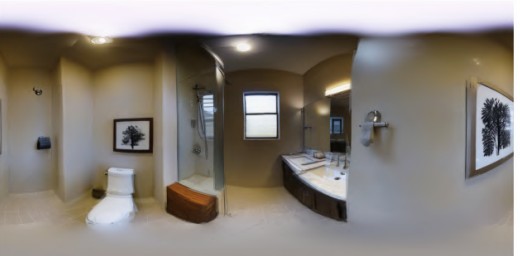 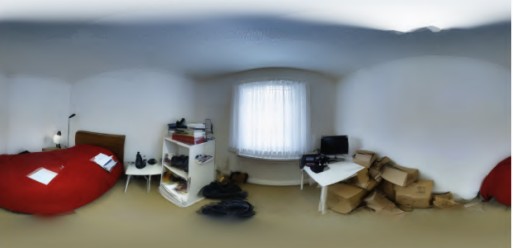

Figure 7: **Qualitative results illustrating the limitations of standard panoramic evaluation metrics**. These images are taken from our model's first-stage outputs, with no refinement stage. There are several noticeable inconsistencies, both in the equatorial and polar regions of these images. However, these images attain an OmniFID score of 45.11, quantitatively outperforming all models, including our full model with the refinement stage. This indicates that OmniFID doesn't fully correlate with panoramic image consistency and quality. Meanwhile, these images have much worse TangentFID and TangentIS scores, only better than StitchDiffusion and MultiDiffusion.

that can be represented without resizing is approximately $1024 \times 512$. In contrast, the tangent-plane formulation allows for significantly higher representational capacity; 18 tangent views with the same resolution already cover up to roughly $1800 \times 900$ in ERP resolution. Moreover, due to the modular nature of the tangent-plane approach, even higher resolutions can be supported simply by increasing the number of tangent views. Nonetheless, to ensure fair comparison with baselines, which are all evaluated at $1024 \times 512$, we standardize our metrics using 18 tangent planes.

Finally, in Figure 7, we provide qualitative comparisons highlighting the failure cases of standard metrics and how TangentIS and TangentFID better correlate with visual quality across the entire panoramic field of view, especially in the polar and boundary regions. Specifically, these results are taken from our first-stage model results, without any refinement. While still a well-formed panoramic image, there are notable inconsistencies in both the polar and equatorial regions of these images. However, these results outperform our own model and all baselines in OmniFID (scoring 45.11 vs our model with refinement, which gets 49.62). This indicates that OmniFID doesn't correlate very well with image consistency in either the polar or equatorial regions. Meanwhile, these results score poorly on both TangentFID and TangentIS (38.73 and 5.46 respectively), only outperforming MultiDiffusion and StitchDiffusion (both of which achieve very poor panoramic results in general).

### A.3.3 ROBUSTNESS TO DISTORTIONS

To highlight the benefits of our proposed metrics over their perspective-based counterparts, we analyze how common panoramic-specific distortions affect evaluation (inspired by the original FID paper, which validated FID using controlled degradations). We consider three types of distortions:

1. Increasing levels of noise localized to polar regions, where ERP distortions are most severe.

2. Edge noise applied at the left and right boundaries to disrupt cycle consistency.

3. Horizontal noise bands of varying intensity across latitudes, reflecting the uneven pixel density of ERP images.

Our results show that both TangentFID and FID reliably increase as noise severity grows, consistent with their intended role as degradation-sensitive metrics. In contrast, the standard Inception Score (IS) behaves counterintuitively, exhibiting a strong positive correlation of $+0.95$ with noise levels under combined distortions, suggesting that spherical artifacts can inflate IS rather than penalize it. By comparison, TangentIS responds as expected, decreasing with a correlation of $-0.99$. This demonstrates that TangentIS more faithfully captures the impact of spherical distortions and provides a perceptually aligned alternative to IS in the panoramic setting.

### A.3.4 USER STUDY

To further validate the perceptual relevance of our proposed metrics, we conducted two user studies designed to compare existing baselines and evaluate how well different metrics align with human judgments.

The first study used 30 paired comparisons between Panfusion (favored by FID/IS) and Diffusion360 (favored by TangentFID/TangentIS). Participants were asked to select the image of higher quality, following the same evaluation protocol described in the main paper. In the second study, we again used 30 pairs, this time comparing our model (TanDiT) against Panfusion, which is preferred by OmniFID.

In the first study, participants overwhelmingly favored Diffusion360 (83% vs. 17%). Importantly, this preference aligns with TangentFID and TangentIS, which rank Diffusion360 higher, but contradicts FID/IS and OmniFID (Table 5)

| Model | FID | IS | OmniFID | TangentFID | TangentIS | Human Preference |
|---|---|---|---|---|---|---|
| Diffusion360 | 42.56 | 3.59 | 48.32 | 37.48 | 5.72 | 83% |
| Panfusion | 35.25 | 4.65 | 48.36 | 39.57 | 5.68 | 17% |

Table 5: **User study 1:** Participants prefer Diffusion360 despite FID/IS ranking Panfusion higher. TangentFID/TangentIS align with human judgments.

In the second user study, human preference was even stronger: TanDiT was chosen in 93% of cases compared to only 7% for Panfusion. Again, this is consistent with TangentFID/TangentIS, but OmniFID shows a negligible difference between the two models (Table 6).

| Model | FID | IS | OmniFID | TangentFID | TangentIS | Human Preference |
|---|---|---|---|---|---|---|
| TanDiT | 31.60 | 4.39 | 47.93 | 32.83 | 5.88 | 93% |
| Panfusion | 35.25 | 4.65 | 48.36 | 39.57 | 5.68 | 7% |

Table 6: **User study 2:** TanDiT is strongly preferred over Panfusion, contradicting OmniFID showing a negligible difference between models, but consistent with the larger gaps given by TangentFID/TangentIS.

Together, these two use studies show that TangentFID and TangentIS correlate more closely with human preference than both perspective-based metrics (FID/IS) and existing panoramic metrics (OmniFID).

## B IMPLEMENTATION AND TRAINING DETAILS

### B.1 DATASET COMPOSITION AND PREPROCESSING

We train our model on a combined dataset composed of panoramic images sourced from Flickr360 (Cao et al., 2023), Polyhaven (polyhaven.com, 2025), and Matterport3D (Chang et al.,

---

**Algorithm 1** Grid Optimization for Tangent-Plane Arrangement

---

**Input:** Tangent view directions $\{(\theta_i, \phi_i)\}_{i=1}^{18}$, Grid size $3 \times 6$
**Output:** Best grid permutation $\pi^*$
  1: Convert each $(\theta_i, \phi_i)$ to 3D unit vector $v_i$
  2: Compute spherical proximity weights $W_{ij}$ between view pairs $(v_i, v_j)$
  3: Define $3 \times 6$ grid and compute pairwise grid distances $D_{pq}^{\text{grid}}$
  4: Initialize permutation $\pi$ via spectral embedding + Hungarian matching
  5: Set best permutation $\pi^* \leftarrow \pi$, best cost $C^* \leftarrow \infty$
  6: **for** $step = 1$ to $N_{\text{steps}}$ **do**
  7:     Propose new permutation $\pi'$ by swapping two grid positions in $\pi$
  8:     Compute cost:

$$C(\pi') = \sum_{i,j} W_{ij} D_{\pi(i),\pi(j)}^{\text{grid}} + \sum_i \left\| (x_i^{\text{erp}}, y_i^{\text{erp}}) - (x_{\pi'(i)}^{\text{grid}}, y_{\pi'(i)}^{\text{grid}}) \right\|_2$$

  9:     **if** $C(\pi') < C(\pi)$ or Accept$(C(\pi'), C(\pi), T)$ **then**
 10:         $\pi \leftarrow \pi'$
 11:         **if** $C(\pi') < C^*$ **then**
 12:             $\pi^* \leftarrow \pi', C^* \leftarrow C(\pi')$
 13:         **end if**
 14:     **end if**
 15:     Update temperature $T$
 16: **end for**
 17: **return** $\pi^*$

---

2017). Specifically, we use 2,700 images from Flickr360, 750 from Polyhaven, and 9,000 from Matterport3D. The panoramas in Flickr360 are provided at a resolution of $1024 \times 2048$, while those from Polyhaven are $2048 \times 4096$. Matterport3D offers panoramic scenes in cubemap format, which we convert into $1024 \times 2048$ equirectangular images as a preprocessing step.

Following the approach of (Li et al., 2022), we extract 18 tangent-plane views from each equirectangular panorama. Due to the distortion properties of equirectangular projection, tangent planes extracted from polar regions exhibit greater distortion and cover a larger area in the scene. To account for this, we extract fewer tangent views from these regions. Specifically, we divide the equirectangular image into four horizontal rows: the first and fourth rows correspond to the north and south poles, from which we extract 3 tangent planes each; the second and third rows are closer to the equator, and we extract 6 tangent planes from each. This results in a total of 18 tangent views, distributed non-uniformly to better match the spatial characteristics of the projection. Each view is downscaled to $192 \times 192$, and arranged into a $3 \times 6$ grid, yielding a final grid image of size $576 \times 1152$, which serves as input to our model during training.

In our experiments, we aim to arrange the tangent planes within a grid such that their placement and adjacency relationships closely approximate their original configuration in the equirectangular panorama. However, because we extract a fixed set of 18 tangent planes, achieving a perfect one-to-one spatial correspondence between grid positions and their true panoramic layout is not possible. To approximate this correspondence as closely as possible, we formulate the grid arrangement as an optimization problem, where a $3 \times 6$ grid is optimized with respect to a grid alignment loss. This loss comprises two components: the first encourages each tangent plane to be positioned near its original panoramic location, while the second enforces consistency in the neighborhood structure by preserving adjacency relationships, aiming to leverage the spatial locality of Diffusion Transformers by keeping overlapping tangent planes close in the grid. Pseudocode of the algorithm used in the grid optimization is shown in Algorithm 1.

A key challenge in panoramic generation is the lack of publicly available datasets with accompanying text captions. To overcome this, we generate dense descriptions using the LLaVA-One-Vision model (Li et al., 2025). However, these captions often exceed the context window of the text encoders used by many baseline models. To ensure fair comparison, we use LLaMA 2 (Touvron et al., 2023) to produce concise summaries, making the captions compatible with varying encoder capacities.

The prompt we provided to LLaVA-One-Vision is

> Give a detailed caption of the following equirectangular projection of a panoramic image. Be detailed about all of the important entities, textures, and colors in the different parts of the scene. Provide enough detail that a text-to-image diffusion model would be able to reconstruct the scene.

and the prompt provided to LLaMA 2 is

> Given a long text prompt that describes a panoramic scene, {longer_caption}, summarize this text prompt to a shorter one that describes what the whole scene looks like without losing important details.

## B.2 MODEL CONFIGURATIONS

**DiT Backbone.** We adopt Stable Diffusion 3.5 Large (SD3) (Esser et al., 2024) as the backbone for our diffusion transformer (DiT) architecture. SD3 incorporates three text encoders—two based on CLIP and one based on a T5 encoder—to support both short and long captions. Input latents are first divided into non-overlapping patches, which are then embedded with positional encodings and processed jointly with the corresponding text embeddings.

Unlike Stable Diffusion 1 and 2, which rely on a U-Net-based noise prediction network, SD3 replaces this component with a transformer-based architecture consisting of 37 Multimodal DiT (MM-DiT) blocks. Each MM-DiT block contains dual processing streams for image latents and text embeddings, enabling the integration of visual and textual modalities. These blocks employ Layer Normalization, Multi-Head Attention, and Feed-Forward layers for expressive and efficient information flow. After the final MM-DiT block, the latent features are projected through a linear layer, unpatchified, and used to predict the denoised output. SD 3.5-Large uses a hidden size of 2432, and a patch size of $2 \times 2$.

**Super Resolution Backbone.** In our experiments, we use VarSR (Qu et al., 2025) as the super resolution model in our panorama generation pipeline. VARSR's pipeline begins by encoding the low-resolution input into prefix tokens that condition the generation process across multiple scales. To preserve spatial structure, VARSR introduces Scale-Aligned Rotary Positional Encoding (SA-RoPE), aligning tokens spatially across scales within an autoregressive transformer. A lightweight diffusion refiner is employed to estimate quantization residuals and recover high-frequency details lost during tokenization. Finally, an image-based classifier-free guidance (CFG) mechanism leverages quality-aware embeddings to guide the generation toward perceptually superior results without additional training.

## B.3 TRAINING PROTOCOLS

We train our model on a combined dataset of 12,450 grid-caption samples, as described in Section B.1, using a batch size of 8. To fine-tune the pre-trained Stable Diffusion 3.5-Large model (Esser et al., 2024), we adopt the LoRA method (Hu et al., 2021), leveraging the AdamW optimizer (Loshchilov & Hutter, 2019) with a learning rate of $1 \times 10^{-4}$ and a constant learning rate scheduler over 10 epochs. Training is conducted on a single Nvidia A40 GPU and takes approximately 40 GPU-hours in total, using the standard rectified flow loss.

## B.4 INFERENCE PROTOCOLS

In the first stage of our panorama generation pipeline, we use the fine-tuned TanDiT model to generate a grid of 18 tangent planes conditioned on a given panorama caption. We employ a diffusion process with 28 timesteps and set the guidance scale to 7.0. To allow for longer captions, we increase the `max_sequence_length` to 512. Inference process of TanDiT is visualized in Figure 8.

In the second stage of Equirectangular-Conditioned Refinement, we perturb the intermediate panorama latent with noise corresponding to a specific timestep. This noisy latent, along with the original caption, is passed to the pre-trained Stable Diffusion 3 model. The model then performs denoising over the specified number of timesteps to produce the final refined panorama.

Timestep ≃ 1000  Timestep ≃ 877  Timestep ≃ 675  Timestep ≃ 515  Timestep ≃ 347  Timestep ≃ 199  Timestep ≃ 0

Figure 8: **Generated panorama by timestep during inference of TanDiT.** TanDit maps random noise to a grid of tangent planes conditioned on a given text.

## C RUNTIME AND MEMORY CONSUMPTION ANALYSIS

TanDiT generates all tangent-plane views simultaneously using a structured grid layout, making the runtime of this stage primarily dependent on the underlying diffusion model. In our experiments, we use Stable Diffusion 3.5 Large (Esser et al., 2024) with 28 inference steps, resulting in a tangent grid generation time of approximately 45 seconds.

Next, the grid is split into 18 individual tangent-plane images, each of which is processed by a pretrained super-resolution model. By default, we use VarSR (Qu et al., 2025) with a $2\times$ upscale factor. The complete super-resolution step for all tangent planes takes approximately 35 seconds.

In the final stage, the upscaled tangent views are reprojected into an equirectangular panorama and encoded into a latent. This latent is then perturbed with noise at a mid-range diffusion timestep and refined using the pretrained DiT model. Our default setting uses 17 denoising steps; however, due to the patched denoising strategy where each timestep involves two passes to maintain spatial consistency, this refinement stage takes up to 55 seconds while refining $1024 \times 2048$ pixels images.

In total, our default pipeline ($2\times$ upscaling with patched refinement) completes in about 135 seconds per image, when we employ Stable Diffusion 3.5 Large as our backbone model. When super-resolution is disabled (thus enabling single-pass denoising), the runtime drops to approximately 85 seconds. Conversely, increasing the upscale factor to $4\times$, which requires 8 patch passes due to increased resolution, raises the total runtime to around 180 seconds. When the smaller Stable Diffusion 3-Medium is employed as the backbone, runtimes decrease to approximately 55, 75, and 100 seconds for generating $512\times1024$, $1024\times2048$ and $2048\times4096$ pixels panoramas, respectively.

For comparison, baseline runtimes range between 11 and 90 seconds (see Table 7). Although our two-stage panorama generation pipeline introduces a modest computational overhead relative to baseline models, it provides significant advantages: the backbone model can be swapped, the super-resolution stage can be disabled if desired, and the approach supports high-resolution fidelity without relying on custom architectures tailored exclusively to panorama generation.

| Method | 512×1024 | 1024×2048 |
|---|---|---|
| MultiDiffusion | 33s | 52s |
| StitchDiffusion | 11s | 21s |
| Diffusion360 | 14s | 1m 31s |
| Panfusion | 44s | – |
| UniPano | 45s | 58s |
| TanDiT (SD3.5-Large) | 1m 25s | 2m 15s |
| TanDiT (SD3-Medium) | 55s | 1m 15s |

Table 7: **Runtime comparison of panorama generation methods at different resolutions.**

When TanDiT employs SD3.5-Large as its backbone, peak memory consumption during inference is approximately 29 GB, which decreases to 16 GB when using the smaller SD3-Medium. For comparison, baseline models require between 3 GB and 21 GB of memory. Although TanDiT exhibits higher memory usage than the baselines, its backbone-agnostic design provides a key advantage: memory requirements scale with the size of the chosen backbone. This is evident from the nearly %50 reduction in memory consumption when switching from SD3.5-Large to SD3-Medium, while maintaining image quality that is on par with or superior to existing baselines. A detailed comparison of memory consumption across models is provided in Table 8.

| Method | Peak Memory Consumption |
|---|---|
| MultiDiffusion | 3 GB |
| StitchDiffusion | 5 GB |
| Diffusion360 | 4 GB |
| Panfusion | 21 GB |
| UniPano | 19 GB |
| TanDiT (SD3.5-Large) | 29 GB |
| TanDiT (SD3-Medium) | 16 GB |

Table 8: **Comparison of peak memory usage during the inference across panorama generation models.**

# D ABLATION STUDIES

## D.1 ABLATING THE POWER OF THE BACKBONE MODEL

TanDiT employs a Diffusion Transformer as the backbone of its pipeline, while remaining model-agnostic in its ability to generate high-quality panoramas independent of the specific underlying model architecture. To validate that TanDiT's performance does not solely arise from the strength of the underlying model, we replace the backbone Stable Diffusion 3.5-Large with the smaller Stable Diffusion 3-Medium, reducing the parameter count from 8.1 billion to 2 billion. Despite this substantial reduction in model capacity, TanDiT continues to achieve performance that is either superior to or competitive with baseline methods across most evaluation metrics. This demonstrates that the quality of the generated panoramas stems primarily from the carefully designed panorama generation pipeline, rather than from the scale of the underlying diffusion model. Quantitative results for TanDiT with Stable Diffusion 3-Medium are presented in Table 9

Table 9: **Quantitative performance comparison of TanDiT with SD3.5-Large and SD3-Medium.** Even when the backbone size is reduced to one quarter (SD3-Medium), TanDiT achieves performance that is on par with or superior to baseline models, further validating the strength and backbone-agnosticity of the carefully crafted panorama generation pipeline.

| Model | FID | KID | IS | CS | FAED | OmniFID | DS | TangentFID | TangentIS |
|---|---|---|---|---|---|---|---|---|---|
| Ours | 31.60 | 0.013 | 4.39 | 24.93 | 1.76 | 47.93 | 0.0003 | 32.83 | 5.88 |
| TanDiT w/ SD3 Medium | 36.28 | 0.016 | 4.74 | 24.69 | 2.83 | 72.35 | 0.0005 | 44.55 | 5.85 |

## D.2 ABLATING THE USAGE OF EQUIRECTANGULAR CONDITIONED REFINEMENT

We further demonstrate the effectiveness of our Equirectangular-Conditioned Refinement strategy by comparing panoramas generated before and after its application. As shown in Figure 9, this refinement step mitigates artifacts and inconsistencies introduced by projecting overlapping tangent planes into the panoramic view. Furthermore, the strong generative capabilities of the pretrained DiT model enhances fine details in the scene, enabling the generation of visually coherent and higher-quality panoramic images.

## D.3 ABLATING THE USAGE OF PATCHED DENOISING IN REFINEMENT STAGE

In our refinement stage, we observe that the underlying DiT model struggles to denoise the upscaled equirectangular panorama at a resolution of $1024 \times 2048$. This leads to noticeable textural degradation, particularly near the polar regions, while the equatorial regions exhibit higher visual quality. We hypothesize that this issue arises because most diffusion transformer models, such as Stable Diffusion 3 (Esser et al., 2024) and Flux (Labs, 2024), are predominantly trained on square-shaped images.

To address this limitation, we divide the panorama into non-overlapping $1024 \times 1024$ pixels square patches, allowing the pretrained DiT model to operate more effectively while still generating rectangular panoramas with a 1:2 aspect ratio. To maintain scene coherence across patch boundaries, we

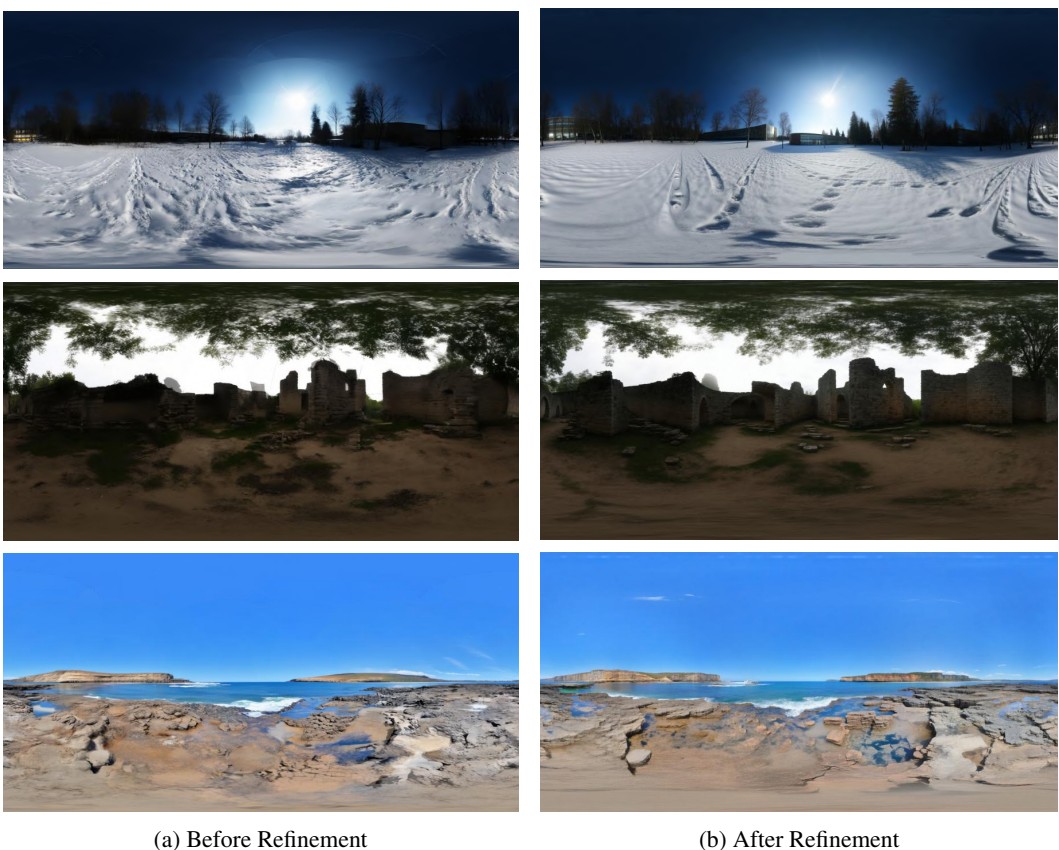

|(a) Before Refinement|(b) After Refinement|

Figure 9: **Example panoramas before and after applying the Equirectangular Conditioned Refinement step.** Refinement step mitigates the artifacts and inconsistencies arising from projecting the tangent planes to a panoramic image, and enables the generation of a visually coherent panorama.

apply circular padding to each patch using pixels from neighboring regions at every denoising step. A qualitative comparison of panoramas refined with and without the patched denoising strategy is provided in Figure 10. Moreover, this strategy facilitates information flow between adjacent patches and enables high-quality generation at higher resolutions, such as 4K ($4096 \times 2048$), as shown in Figure 14.

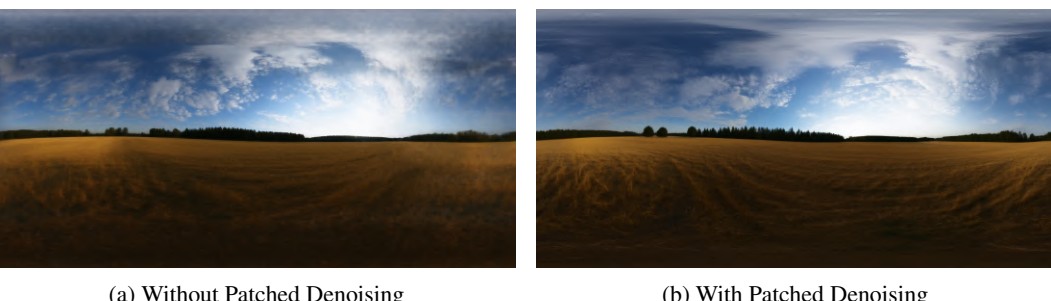

|(a) Without Patched Denoising|(b) With Patched Denoising|

Figure 10: **Effect of patched denoising on high-resolution panoramic generation.** (a) Without patched denoising, the model struggles to maintain texture quality and consistency across the image, resulting in visible artifacts and degradation near the boundaries. (b) With patched denoising, the image retains sharper details, improved coherence, and better visual quality across the entire panoramic scene, demonstrating the effectiveness of this strategy in high-resolution settings.

### D.4 ABLATING LATENT ROTATION AND CIRCULAR PADDING IN PATCHED DENOISING

Our patched denoising strategy inherently introduces potential inconsistencies between patches, as we operate on non-overlapping square regions independently. To promote information sharing across neighboring patches, we apply circular padding in each denoising step and again before VAE decoding. Specifically, each patch is padded using pixels from its adjacent patches, allowing the model to denoise patches separately while maintaining global scene coherence.

However, we observe that this approach can still introduce seams in the interior regions of the patches. This occurs because the model primarily focuses on aligning the boundaries between patches, while potentially neglecting internal consistency within each patch. To address this, we adopt a latent rotation strategy: during each denoising step, we cyclically rotate the entire latent representation along the horizontal axis. This encourages the model to attend to different spatial regions throughout the denoising process, rather than focusing on a fixed area. After denoising, we reverse the rotation to restore the original spatial alignment and ensure that the generated panorama remains consistent with the input prompt in terms of content layout and position. The combined effect of circular padding and latent rotation is illustrated in Figure 11.

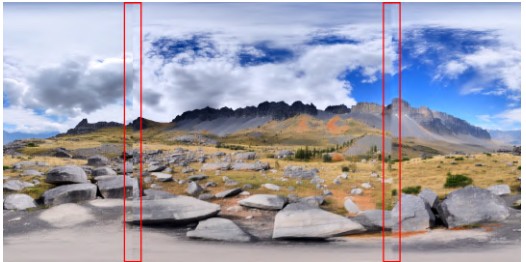 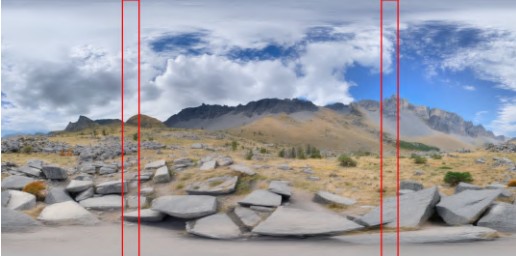

(a) Without Latent Rotation and Circular Padding      (b) With Latent Rotation and Circular Padding

Figure 11: **Effect of latent rotation and circular padding in patched denoising.** Images are horizontally rotated by 90° to better visualize the loop-consistency. Original left-right loop regions are marked with red. (a) Without circular padding, there is no sufficient information flow between the left and right edges of the image during the denoising. Thus, the model is prone to generating panoramic images that are not loop-consistent. (b) With circular padding combined with latent rotation, the model is able to denoise the patches while keeping the whole scene visually coherent.

### D.5 ABLATING THE NOISE LEVEL USED IN REFINEMENT STAGE

To justify our choice of $T_{\text{high}} \simeq 700$ which corresponds to 13 steps of inference, we conduct ablation experiments with two alternative settings: $T_{\text{high}} \simeq 600$ (10 steps) and $T_{\text{high}} \simeq 800$ (17 steps). The results are presented in Table 10. We observe that using a higher timestep, $T_{\text{high}} \simeq 800$, improves generic image quality metrics such as Inception Score (IS) and CLIP Score (CS), likely due to the model having more flexibility to reshape fine details. However, this comes at the cost of degrading panorama-specific metrics (FAED, OmniFID, DS, TangentFID, TangentIS), as more of the original scene structure is lost during denoising. Conversely, using a lower timestep, $T_{\text{high}} \simeq 600$, better preserves the original panoramic structure, resulting in higher scores on metrics tailored to global consistency. That said, it also leaves more residual artifacts and inconsistencies due to insufficient refinement, reflected in reduced IS and CS, and increased FID and KID metrics.

This trade-off is visually illustrated in Figure 12, which shows the output of our refinement process using each of the tested timestep settings. Lower timesteps primarily reduce high-frequency noise, while higher timesteps yield stronger visual changes that may alter prompt-relevant details. Our default choice of $T_{\text{high}} \simeq 700$ allows for a balance between these two extremes, preserving essential panoramic content while allowing sufficient refinement to improve visual quality and semantic alignment.

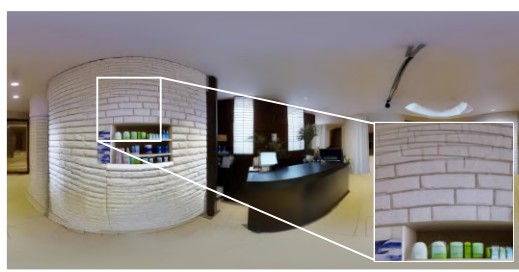 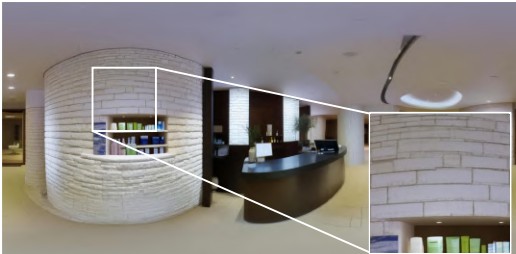

| (a) Initial Generation Result | (b) Refinement with $T \simeq 600$ |

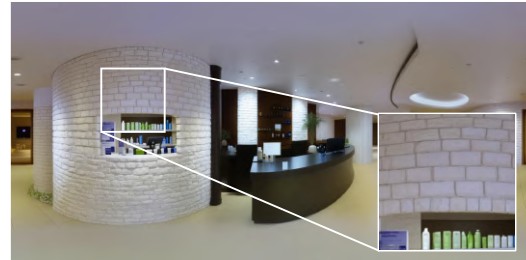 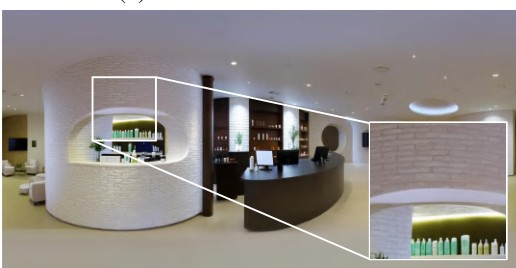

| (c) Refinement with $T \simeq 700$ | (d) Refinement with $T \simeq 800$ |

Figure 12: **Effect of timestep in the Equirectangular-Conditioned Refinement process.** We visualize the effect of different noise levels used during refinement in TanDiT. Starting from the initial equirectangular panorama, we apply refinement using noise injected at various timesteps ($\simeq 600$, $\simeq 700$, $\simeq 800$). Lower timesteps yield subtle improvements, primarily reducing high-frequency artifacts, while higher timesteps introduce more aggressive changes that may alter fine-grained content. As shown in the insets, refinement improves visual coherence and detail consistency in overlapping regions. In our final model, we adopt a timestep of $\simeq 700$ as a balance between visual fidelity and effective artifact removal.

Table 10: **Comparison of different timesteps for our equirectangular-conditioned refinement process.** Picking the number of timesteps for refinement is crucial for balancing the quality of the refinement, while still preserving the crucial characteristic structure of the equirectangular projected panoramic image. Going too low, (600), preserves more structure but fails to refine the images enough, so the quality is lower and equirectangular projection artifacts are visible. Meanwhile, going too high, (800), destroys a lot of the panoramic structure, but allows for higher-quality images. A timestep of (700) allows for balancing these two to obtain images with superior quality.

| Num Timesteps | FID | KID | IS | CS | FAED | OmniFID | DS | TangentFID | TangentIS |
|---|---|---|---|---|---|---|---|---|---|
| Ours ($\simeq 700$) | **31.60** | **0.013** | 4.39 | 24.93 | 1.76 | 47.93 | **0.0003** | 32.83 | 5.88 |
| ($\simeq 600$) | 32.35 | 0.014 | 4.30 | 24.67 | **1.69** | **44.22** | **0.0003** | **33.19** | **5.93** |
| ($\simeq 800$) | 32.93 | **0.013** | **4.50** | **25.33** | 1.84 | 53.84 | 0.0004 | 35.10 | 5.85 |

## D.6 PRETRAINED STABLE DIFFUSION WITH OUR CAPTIONS

To validate that the performance gains achieved by our method stem from both the proposed training procedure and post-processing steps, and are not merely a result of using a stronger backbone model (Stable Diffusion 3), we conduct a control experiment. Specifically, we provide the same captions to the default Stable Diffusion 3 model, without any LoRA fine-tuning or refinement steps, and evaluate its outputs using the same metrics. To further assist the baseline model, we modify the caption with the phrase "A panoramic image of ...". The results are summarized in Table 11. As shown, the outputs from the unmodified baseline are significantly worse across all key metrics (with the exception of Inception and CLIP scores, whose limitations we have already discussed in the main paper).

Table 11: **Comparison between our proposed method (TanDiT) and the pretrained SD3 model.** We evaluate the default SD3 model without fine-tuning or post-processing, with and without the trigger phrase "A panoramic image of ...". TanDiT outperforms both variants across all key panoramic-specific metrics, confirming that our gains are due to the proposed training and refinement pipeline, not just the underlying backbone. Although SD3 achieves higher IS and CLIP scores, these are less reliable for panoramic evaluation, as discussed in the main paper.

| Model | FID | KID | IS | CS | FAED | OmniFID | DS | TangentFID | TangentIS |
|---|---|---|---|---|---|---|---|---|---|
| TanDiT | **31.60** | **0.013** | 4.38 | 24.93 | **1.76** | **47.93** | **0.0003** | **32.83** | **5.88** |
| Pretrained SD3 | 67.63 | 0.041 | **6.29** | **25.89** | 4.29 | 79.13 | 0.0050 | 63.16 | 2.50 |
| + Trigger Phrase | 48.12 | 0.023 | 5.09 | 24.96 | 4.63 | 69.97 | 0.0039 | 51.64 | 2.55 |

Table 12: **Effect of caption length on model performance.** We evaluate the impact of long vs. summarized captions on both TanDiT and the baselines. TanDiT is trained using summarized captions, while the baselines are trained with full, detailed captions (truncated based on each model's token limit). Despite this reversal in caption advantage, TanDiT continues to outperform all baselines across all panoramic-specific metrics, confirming that its superior performance is not solely due to access to longer text inputs.

| Model | FID | KID | IS | CS | FAED | OmniFID | DS | TangentFID | TangentIS |
|---|---|---|---|---|---|---|---|---|---|
| TanDiT | **33.19** | **0.012** | 4.49 | **25.92** | **2.94** | **55.71** | **0.0004** | **39.96** | 5.38 |
| UniPano | 46.77 | 0.028 | 4.47 | 19.7 | 5.12 | 68.70 | **0.0004** | 53.31 | 5.25 |
| Panfusion | 34.60 | 0.023 | 4.71 | 20.65 | 3.55 | 57.86 | 0.0005 | 47.01 | **5.60** |
| StitchDiffusion | 77.53 | 0.053 | 4.74 | 19.78 | 9.90 | 115.29 | 0.0015 | 69.23 | 2.45 |
| Diffusion360 | 48.42 | 0.028 | 4.04 | 22.11 | 4.37 | 55.79 | 0.0006 | 46.29 | 2.48 |
| MultiDiffusion | 73.16 | 0.044 | **7.27** | 23.95 | 4.31 | 87.78 | 0.0032 | 61.76 | 2.42 |

### D.7 ABLATING THE IMPACT OF OUR CAPTIONS ON THE RESULTS

Stable Diffusion 3 can handle substantially longer captions than earlier models, up to 512 tokens, compared to just 77 in Stable Diffusion 1 and 2. As discussed previously, we used summarized captions for training the baseline models and for computing the CLIP score, to ensure compatibility with their shorter context lengths. To verify that our model's improved performance is not simply due to access to longer, more detailed captions, we conduct two additional experiments: (1) we train our proposed method (TanDiT) using the summarized captions, and (2) we train all baseline models using the full, detailed captions (noting that these will be truncated when exceeding each model's token limit). The results, presented in Table 12, confirm that the performance gap cannot be attributed solely to caption length.

### D.8 ABLATING OUR ORDERING OF THE GRID

While we adopt a specific optimized grid ordering as explained in Section B.1 and as shown in Figure 13(c), there exist numerous alternative arrangements that can maintain a degree of local consistency between neighboring tangent planes.

We evaluate our chosen configuration against two alternatives: (1) a row-wise ordering, where tangent planes are arranged left-to-right, top-to-bottom without adjustment. This would correspond to relocating the three top-right tangent planes to the bottom (Figure 13(b)), and (2) a column-wise ordering, proceeding top-to-bottom, left-to-right (Figure 13(d)).

For perspective image-based metrics (FID, KID, IS, CS), all three configurations yield comparable results. However, our proposed ordering achieves superior performance on panorama-specific metrics (OmniFID, TangentFID, TangentIS), which motivated its selection. Quantitative results comparing these grid arrangements are presented in Table 13.

Table 13: **Comparison of different orderings for our grid layout.** We can see that while the perspective image metrics are similar, our layout performs significantly better than the rest on the panoramic image metrics.

| Model | FID | KID | IS | CS | FAED | OmniFID | DS | TangentFID | TangentIS |
|---|---|---|---|---|---|---|---|---|---|
| Ours | **31.60** | 0.013 | 4.39 | **24.93** | 1.76 | **47.93** | **0.0003** | **32.83** | **5.88** |
| Row-wise | 32.40 | **0.012** | **4.52** | 23.55 | 4.03 | 52.91 | 0.0004 | 36.58 | 5.47 |
| Column-wise | 31.74 | **0.012** | 4.30 | 23.63 | **1.67** | 54.36 | 0.0004 | 36.41 | 5.40 |

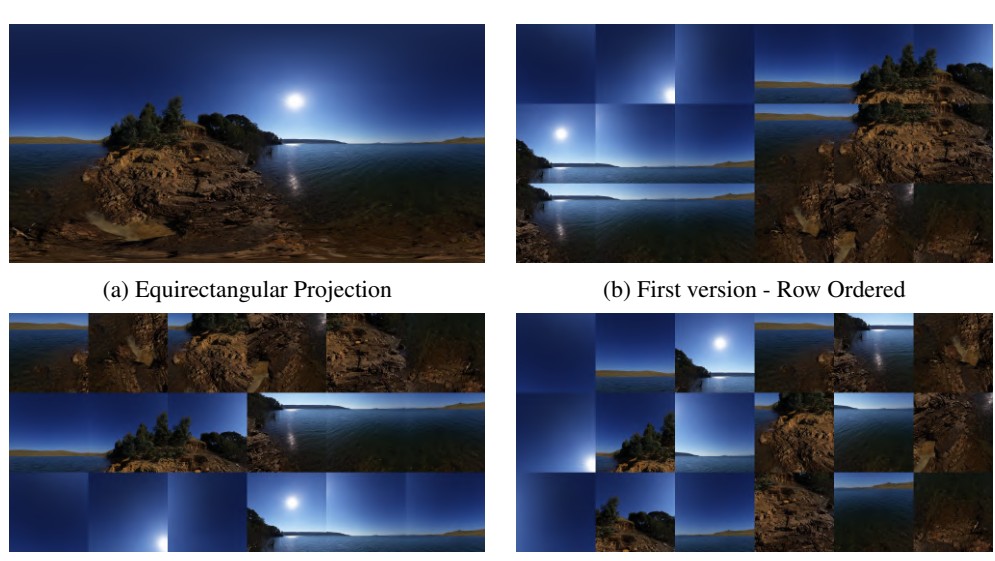

(a) Equirectangular Projection

(b) First version - Row Ordered

(c) Second version - Our optimized ordering

(d) Third version - Column Ordered

Figure 13: **A comparison of panorama representations.** (a) Tangent planes are arranged row-wise from top to bottom. (b) Tangent planes are arranged according to an optimized grid arrangement that aims to minimize the displacement of the tangent planes and the distance between the neighbouring tangent planes. (c) Tangent planes are arranged column-wise from left to right.

### D.9 ABLATING NUMBER OF TANGENT PLANES IN THE GRID

To evaluate the effect of the number of tangent planes on model performance, we experiment with generating 10 and 26 tangent planes per scene, in addition to our default setting of 18. The results of the metrics are shown in Table 14. In the case of 26 tangent planes, we observe that the underlying DiT model struggles to learn the grid structure effectively, likely due to the increased complexity introduced by a denser layout. In contrast, both the 10 and 18 image configurations result in successful grid learning and the generation of visually coherent panoramas.

However, using a higher number of tangent planes offers advantages for high-resolution panorama generation. Since each tangent plane can be independently passed through a super-resolution pipeline, increasing their count allows the final equirectangular panorama to achieve higher resolution without requiring aggressive upscaling. This reduces the risk of introducing artifacts, as more native-resolution tangent views are available to cover the output equirectangular projected space.

### D.10 ABLATING THE USE OF SUPER-RESOLUTION

Figure 14 demonstrates that TanDiT can effectively generate high-quality panoramas at resolutions up to 4K. To quantify the impact of super-resolution in achieving this, we perform an ablation comparing results with and without applying super-resolution to the generated tangent-plane images. Applying super-resolution consistently improves performance across most metrics, especially those sensitive to texture and detail (e.g., FID, TangentFID, FAED). By employing our de-

Table 14: **Comparison of our model with varying levels of tangent planes in the grid.** We see that our choice of 18 tangent planes outperforms both other choices in the panoramic-specific metrics. Using 10 planes does lead to higher distortion per plane, affecting the final panoramic results. Meanwhile, the model struggles to properly learn the grid structure in the case of 26 tangent planes, leading to suboptimal results.

| Num Planes | FID | KID | IS | CS | FAED | OmniFID | DS | TangentFID | TangentIS |
|---|---|---|---|---|---|---|---|---|---|
| Ours (18) | **31.60** | 0.013 | 4.39 | **24.93** | **1.76** | **47.93** | **0.0003** | **32.83** | **5.88** |
| 10 | 32.04 | **0.012** | 4.50 | 23.52 | 1.78 | 53.12 | 0.0005 | 36.10 | 5.53 |
| 26 | 42.81 | 0.019 | **4.61** | 23.01 | 6.34 | 77.58 | 0.0005 | 47.64 | 5.04 |

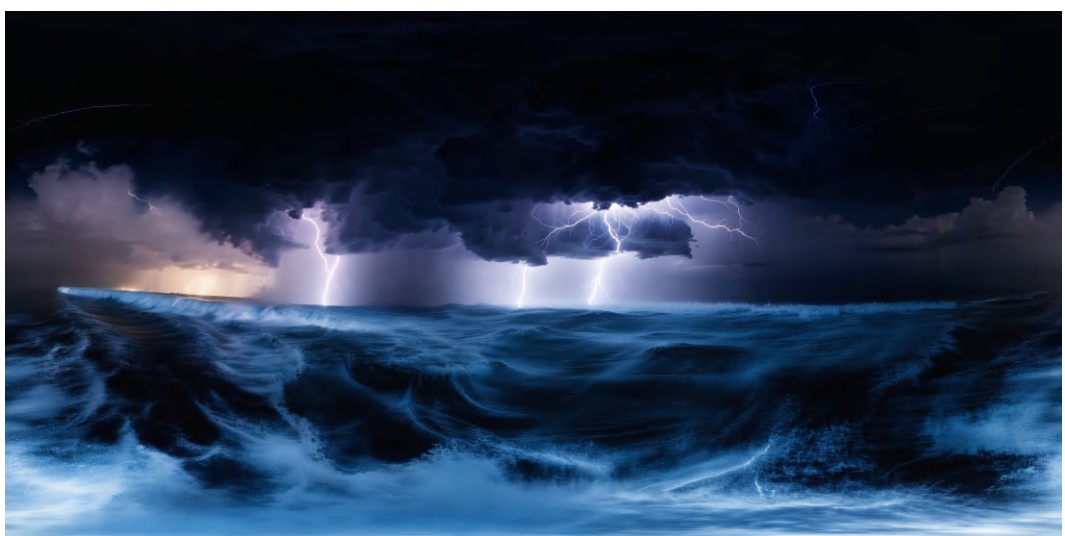

Figure 14: **A 4k image generated by TanDiT.** Employing the proposed patched denoising technique in the Equirectangular-Conditioned Refinement step, we are able to generate 4k images.

fault 2× super-resolution strategy, TanDiT enhances spatial fidelity within each tangent plane prior to stitching and refinement. Additionally, integrating super-resolution with patched denoising in the Equirectangular-Conditioned Refinement stage provides flexible control over output resolutions, such as producing detailed 2K or 4K panoramas.

# E  ADDITIONAL QUALITATIVE RESULTS

## E.1  GENERATING OUT-OF-DOMAIN STYLIZED PANORAMAS WITH TANDIT

Leveraging the powerful generative capabilities of the underlying DiT model within the Equirectangular-Conditioned Refinement stage, TanDiT is capable of synthesizing high-quality stylized panoramas well beyond its original training distribution. By conditioning on descriptive textual prompts, TanDiT successfully produces panoramas exhibiting diverse artistic styles, such as "Pixel Art", "Watercolor Painting" and "Monochrome". This flexibility enables users to explore a wide variety of visually distinct aesthetics, significantly broadening TanDiT's potential applications, from digital art creation to immersive media content.

Figure 15 provides additional visual examples demonstrating TanDiT's ability to generalize across these out-of-domain styles. Notably, these results illustrate that TanDiT maintains spatial coherence and panoramic consistency even when generating highly abstract or stylistically complex imagery. Future work could further investigate the integration of style-specific fine-tuning or few-shot learning to enhance quality and control in specialized stylistic domains.

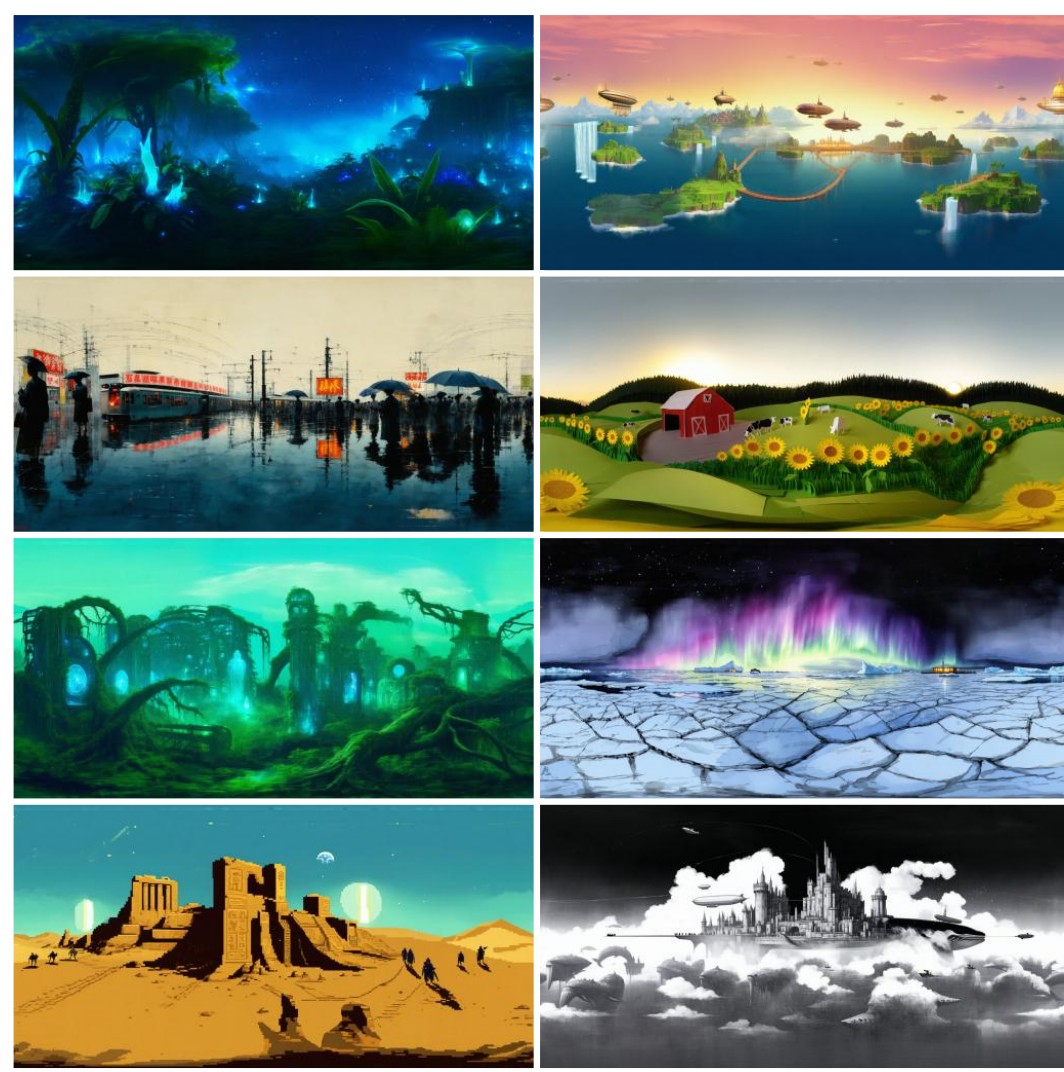

Figure 15: **Out-of-domain stylized panoramas generated by TanDiT.** TanDiT, combined with the Equirectangular-Conditioned Refinement method is able to generalize to many out-of-domain styles.

### E.2 QUALITATIVE COMPARISON WITH CUBEDIFF

CubeDiff (Kalischek et al., 2025) is a panoramic outpainting model that takes a perspective image and a text caption as input and generates a panoramic output by producing a cubemap, with the input perspective image used as one of its faces. Although the code is not publicly available, we provide qualitative comparisons based on the results presented on their project webpage, highlighting how our method avoids the consistency issues observed in CubeDiff.

CubeDiff generates each cubemap face independently, applying modified attention layers to enable limited interaction between the faces. The final panorama is obtained by converting the cubemap into an equirectangular projection. While this attention mechanism introduces some amount of coherence over cube faces, it does not fully resolve inconsistencies at the seams. Notably, distortions inherent to the cubemap projection, stronger near the face boundaries than at the center, are not appropriately modeled. Since CubeDiff treats each face as an undistorted perspective image, this results in visual artifacts when the faces are stitched together, particularly in regions with high distortion near the cube edges. See Section A.3 for a further discussion about the distortion produced by cubemaps near the edges. For example, Figure 16 shows a prominent seam at the center of

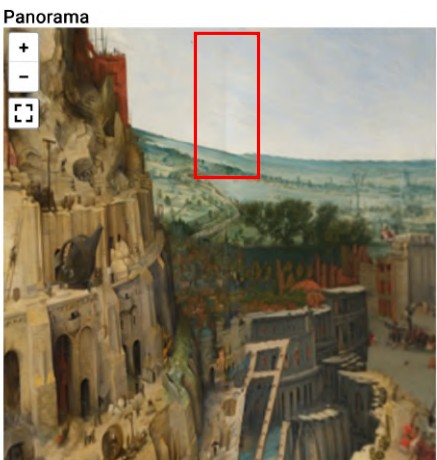

Figure 16: **Example of inconsistency in CubeDiff's panorama generation.** The left image shows the perspective input used to condition CubeDiff. The right image is the resulting equirectangular panorama, where a visible seam appears near the center due to misalignment and distortion from cubemap face warping. This illustrates a common failure mode in CubeDiff, where treating each cubemap face as an undistorted perspective view leads to artifacts at face boundaries. Best viewed zoomed-in.

the image, caused by misalignment and warping during cubemap conversion. Although this is a particularly clear case, similar issues can be seen in many other results on their website. In some indoor scenes, seams may be harder to detect due to symmetric layouts (e.g., four-wall rooms), but inconsistencies still persist.

## F  LOOP-CONSISTENCY ANALYSIS

To support our claim that the generated panoramas exhibit strong loop-consistency, we provide qualitative visualizations under various settings. For improved clarity, each panorama is rotated by 90 degrees, allowing the left and right boundaries to be shown side by side. The boundary regions, where seamless continuity is most critical, are highlighted with red rectangles in Figure 17. These visualizations demonstrate that our refinement strategy, combined with circular padding, enables smooth transitions across the horizontal edges of the panorama.

## G  USER STUDY DETAILS

As described in Section 5.2 of the main paper, we conducted a user study via Qualtrics to visually assess the effectiveness of our method. Participants were presented with paired comparisons between our model's output and that of a baseline model, and asked to select the image they preferred based on a given prompt. A total of 30 volunteers participated in the study and were provided with the following evaluation criteria:

---

You'll review panoramic images generated from text prompts. Please rate each image based on the following:

**Relevance**: Does the image faithfully represent the prompt? Are key elements included?

**Realism**: Does it look natural and believable? Are there any distortions or artifacts, especially near seams or poles?

**Seam Consistency**: Are the left and right edges smoothly connected? Is spatial consistency maintained?

**Coverage**: Is the scene immersive and complete, without blank areas or awkward repetitions?

**Overall Impression**: How would you rate the image's quality as a whole?

---

Figure 18 shows an example question from the user interface. Participants compared two videos rendered from panoramic images with identical camera settings, resolution, and rotation speed. For each question, one image was generated by our method and the other by a randomly selected baseline. The left/right positions were randomized to avoid positional bias.

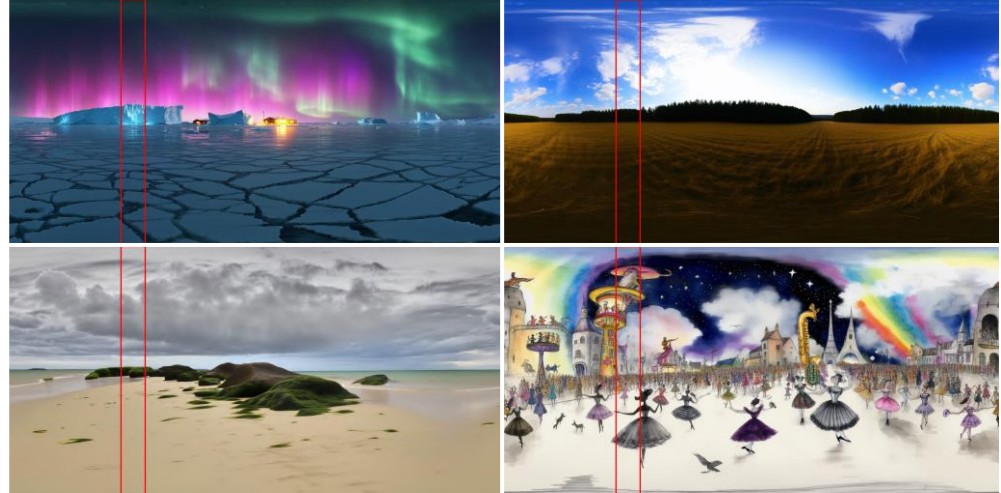

Figure 17: **Loop Consistency in the refined panoramas.** Refined panorama images are rotated by 90 degrees horizontally and marked with a red rectangle to qualitatively show the left-right continuity. Utilizing circular padding in the refinement step allows for a sufficient information flow in the left and right edges, thus enabling the generation of a fully loop-consistent panorama.

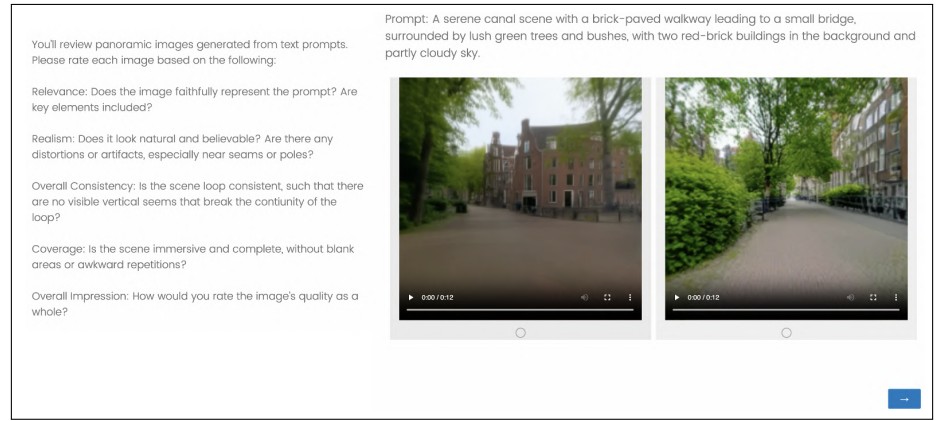

Figure 18: **Screenshot of the survey layout.** The participants are asked to choose the best generation result with high relevance to the text prompt, realism, seam consistency, coverage, and overall impression.

Figure 19 summarizes the user preferences. Across all baselines, our method is consistently preferred, with a 95% Wilson confidence interval confirming statistical significance in every case. These results indicate that our approach better satisfies the key visual quality criteria compared to existing methods. Note that we discovered the UniPano Ni et al. (2025a) model after the user study had been completed, so we were unable to include it in the results.

## H FURTHER LIMITATIONS

In this section, we expand on the technical and ethical limitations of our approach, complementing the discussion provided in the main paper.

As mentioned in the conclusion, our pipeline relies on two versions of the SD3 model: the first stage uses a LoRA-fine-tuned SD3 for tangent-plane generation, while the refinement stage requires the original, unmodified SD3. Consequently, both sets of weights must be available during inference.

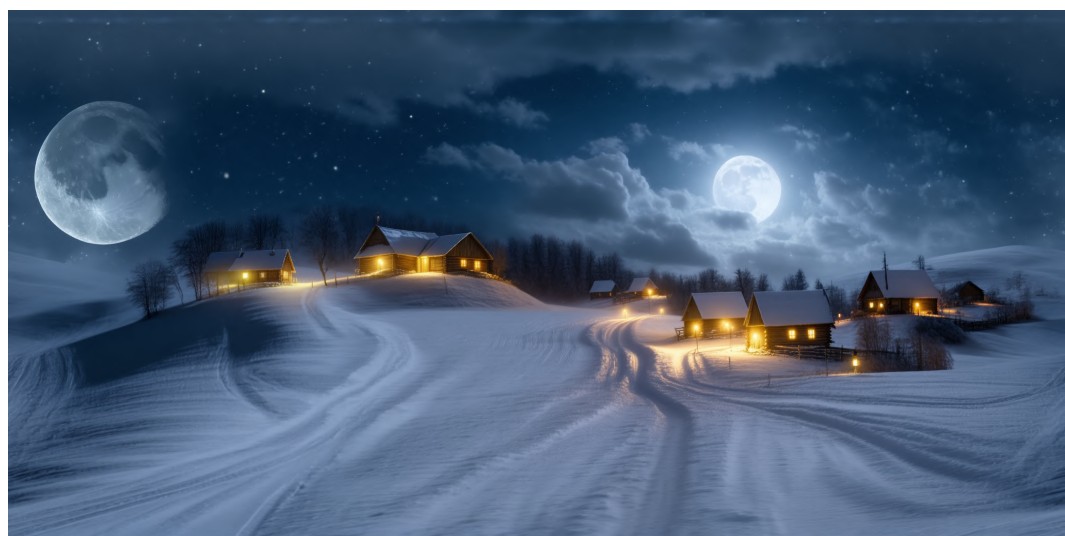

Figure 19: **Results of the pairwise user preference study comparing our method against five baselines**. Each point shows the percentage of times our approach was preferred over a given baseline, along with a 95% Wilson confidence interval. Across all comparisons, our method is significantly preferred, indicating that it better satisfies key qualitative criteria such as realism, relevance to the prompt, seam consistency, and overall visual quality.

Figure 20: **A limitation of the patched denoising approach.** In this example, two patches independently generated a moon, resulting in duplicate content in the final image.

Although our tangent-plane representation reduces the geometric distortions commonly found in equirectangular and cubemap projections, it does not eliminate them entirely, especially near the poles. In theory, increasing the number of tangent views would yield more accurate reconstructions, but doing so would introduce significant memory and computational overhead. Moreover, current generative models like SD3 remain limited in their ability to produce high-resolution content at large scale.

To support high-resolution output (e.g., 4K panoramas), we apply a patched denoising strategy in the refinement stage, where the latent representation of the ERP image is split into smaller segments, denoised independently, and then stitched back together. While this improves scalability, it can lead to duplicate content, as each patch is conditioned on the same global caption without local awareness. As illustrated in Figure 20, this may result in artifacts such as repeated objects, e.g. two moons generated in different patches of the sky. While the panorama remains globally consistent and structurally valid, such repetitions may deviate from the intended semantics of the prompt.

The refinement stage also remains essential for correcting inconsistencies introduced by the tangent-plane generation process. These artifacts result from the flow-matching loss used during training,

which does not impose explicit geometric or pixel-wise consistency. While alternative strategies such as stronger geometric loss functions, alignment-aware objectives, or inference-time guidance could address these issues, we leave such explorations to future work.

Lastly, we acknowledge potential ethical concerns. As with other generative models, TanDiT could be misused to produce photorealistic but fictitious 360° content, potentially contributing to misinformation or deceptive media, especially in immersive applications like VR. Although this risk is not unique to our method, it highlights the importance of responsible deployment, including moderation tools, usage guidelines, and methods such as watermarking to detect synthetic content.

## I LLM USAGE

LLMs were used for grammatical checking and for improving the flow and clarity of the text. All final revisions were performed by the authors.

## J LICENSE, DATASET, AND BENCHMARK RELEASE PLANS

Upon acceptance, we will publicly release the following assets to support reproducibility and foster further research in panoramic image generation:

- Both full and summarized captions for the Flickr360, Polyhaven, and Matterport3D datasets used in our experiments.

- Precomputed tangent-plane grids (in both image and latent formats) for all training and evaluation samples.

- A unified implementation of all evaluation metrics, including standard metrics (FID, IS, CLIP Score, OmniFID, FAED, DS) and our newly proposed TangentFID and TangentIS.

- Complete code for data preprocessing, training, inference, refinement, evaluation, and visualization, along with instructions and configuration files.

- The LoRA-fine-tuned Stable Diffusion 3.5-Large model used in our experiments, along with all necessary configuration files.

All assets will be hosted on GitHub (for code and documentation) and HuggingFace (for datasets and model weights). Captions and code will be released under a CC-BY-NC 4.0 license. Model weights and training scripts will follow the original license of Stable Diffusion 3.5 and HuggingFace's model hub guidelines.

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
