# OpenReview forum: "Geometry-Aligned Tangent-Plane Diffusion Transformers for 360° Panorama Generation"
_ICLR.cc/2026/Conference — ICLR 2026 Conference Withdrawn Submission_

### Official Review · Reviewer_BwcN · 2025-10-29

**Soundness:** 2
**Presentation:** 2
**Contribution:** 2
**Rating:** 2
**Confidence:** 4

**Summary:**

This paper proposes TanDiT for text-to-360° panorama generation. The core idea is to factorize the sphere into tangent-plane views, arrange them as a 2D grid, and fine-tune a pretrained DiT (SD3) on this grid with conditional flow matching. A refinement stage reprojects the grid to ERP and performs SDEdit-style partial denoising with circular padding to suppress seams. The authors also introduce TangentFID/TangentIS, which compute FID/IS over tangent patches and report a 95% confidence bound to emphasize worst-region quality. On several datasets, TanDiT reports strong scores on standard and spherical metrics and includes ablations/user studies supporting its design.

**Strengths:**

- Building on a large SD3 DiT and fine-tuning on tangent grids enables global coherence across views in one diffusion loop, a practical simplification compared to multi-step/tile methods. The approach benefits from long-range attention and pretraining, yielding high visual quality.

- The method reports SOTA or near-SOTA across standard and panoramic metrics and includes ablation studies showing the importance of circular padding, patch-wise high-aspect-ratio denoising, latent rotation, and SR. This supports that each module contributes meaningfully to seam reduction and fidelity.

**Weaknesses:**

- The pipeline concatenates views into a single grid and then solves a nontrivial permutation problem to enforce attention locality. By contrast, prior multi-view image generation works [1], [2], [3] treat each view independently and apply a shared cross-view attention to model the joint distribution in an order-invariant set formulation; although the authors claim an advantage in reusing an off-the-shelf DiT without special spherical modules, fine-tuning is required regardless, and the justification for choosing grid-concatenation instead of established multi-view attention is unclear. The paper provides neither a theoretical justification nor controlled empirical evidence showing that this design is superior to order-invariant multi-view attention.

- The refinement stage applies SDEdit-style noising/denoising to ERP latents using the SD3 backbone (without ERP-specific fine-tuning). SDEdit’s rationale typically presumes the clean target lies on (or near) the denoiser’s learned data manifold; however, SD3 was not trained on ERP/panoramic images. The paper does not establish that ERP panoramas lie on SD3’s manifold or provide empirical analyses, so the theoretical basis for this refinement step remains weak despite the observed seam reduction.

- The paper asserts (Sec. E.2, Sec. A.3) that cubemap edge distortion explains CubeDiff’s seam artifacts. I am not convinced by this claim. Under the pinhole camera model, all perspective views exhibit the same class of projection “distortions”. A cubemap face (≈90° FOV) is itself a standard pinhole-perspective view of the scene, no different in kind from ordinary camera photographs. The term “regular perspective image (without distortion)” is therefore ill-defined/misleading in this context. A more plausible explanation for the mild seams observed in CubeDiff is data scarcity or FOV mismatch between the conditioning image and the ≈90° cubemap faces. As written, the paper attributes a generic property of pinhole projection to a cubemap-specific failure mode; this claim should be corrected or substantially qualified.

[1] Tang, S., Zhang, F., Chen, J., Wang, P., & Furukawa, Y. (2023). MVDiffusion: Enabling Holistic Multi-view Image Generation with Correspondence-Aware Diffusion. arXiv.

[2] Shi, Y., Wang, P., Ye, J., Mai, L., Li, K., & Yang, X. (2023). MVDream: Multi-view Diffusion for 3D Generation. arXiv:2308.16512.

[3] Huang, Z., Guo, Y., Wang, H., Yi, R., Ma, L., Cao, Y.P., & Sheng, L. (2024). MV-Adapter: Multi-view Consistent Image Generation Made Easy. arXiv preprint arXiv:2412.03632.

**Questions:**

- Your metrics compute per-view (tangent-patch) scores and then report a 95% confidence bound (lower for IS, upper for FID). This design may favor panoramas that are uniformly average over ones that are excellent overall but contain a single localized artifact, making the metric potentially sensitive to outliers. Could you provide the per-patch mean and standard deviation for all baselines alongside the reported confidence bounds?

While the paper leverages a modern DiT backbone to generate visually impressive $360^\circ$ panoramas, the evidential support for its main contributions is not convincing. Key design choices—such as grid-concatenation vs. order-invariant multi-view attention, the attribution of CubeDiff seams to cubemap distortion, the design of the TangentFID/TangentIS metrics that emphasize worst-case regions, and the ERP-space refinement using an SD3 prior—remain under-justified and insufficiently validated. Consequently, I do not believe the submission meets the standards of this conference.

---

### Official Review · Reviewer_4tuo · 2025-10-29

**Soundness:** 3
**Presentation:** 3
**Contribution:** 3
**Rating:** 4
**Confidence:** 4

**Summary:**

The work introduces TanDiT, a "tangent-plane" diffusion model for 360° panorama generation. TanDiT argues that proper representation is the gap in generating high-quality 360° panoramas, and proposes using a factorized representation of the unit sphere into planar patches, enabling high-quality spherical panorama generation without architectural changes. After generation, using the "ERP-conditioned refinement", patches are refined/harmonized to resolve overlap and seams artifacts and improve global consistency. The paper also introduces two new metrics that are more aligned for 360° panoramas, TanFID and TanIS, which capture degradations at poles and seams (common sources of artifacts) and better align with human perceptual preference.

**Strengths:**

- The results appears compelling, both qualitatively and quantitatively, and show good human preference as well.
- The method is straightforward and easy to understand.
- An elegant solution and good empirical execution.
- The paper is well written and easy to follow.

**Weaknesses:**

- **Ablating the ERP-conditioned refinement stage.** Given that the tangent-plane representation is stated as being a primary contribution and central component to the paper's argument, it is important to properly isolate it's value.
    - D.2 shows qualitative results for the ERP-conditioned refinement, however quantitative results are missing.
    - Cubemaps can be thought of a subset of tangent-plane projections where $|\mathcal{G}| = 6$ and grid layout is, naturally, defined along the rules of a cubemap. Experiments showing the refinement stage applied to ERP-based methods is missing.
- Missing qualitative results for MultiDiffusion
- Comparison to SphereDiff is missing
    - Given the overlap in novelty, this is important.
    - This paper improves over SphereDiff in computational cost, and therefore that brings merit to its novelty, however this comparison is still important.
- (minor) Comparison to U-Net architectures would be beneficial, and support the claim the DiT model spatial correlations more effectively.
- (minor) Outlining artifacts in the panoramas would be visually useful to improve digestion of the paper. For example in Figure 9, showing the before and after, it is very difficult to see what/where improvements are made.
- (minor) L147 typo, duplicate wording.
- (minor) Section 3, a graphical representation of the tangent-plane projection could go a long way to help readers internalize the representation.

**Questions:**

- What is the 1/2 generation in Table 1? At first this reads as "0.5" - later it is more clear that this may mean "1 or 2" (including the refinement stage?)
- What are the known failure modes of TanDiT, especially in challenging scenes or out-of-domain prompts?
- Are there cases where the refinement stage introduces new artifacts, such as duplicated objects or loss of semantic detail?

---

### Official Review · Reviewer_ypY3 · 2025-11-01

**Soundness:** 2
**Presentation:** 2
**Contribution:** 1
**Rating:** 4
**Confidence:** 5

**Summary:**

The paper presents TanDiT, a tangent-plane diffusion transformer for 360$\degree$ panoramic image generation. Previous approaches using ERP and cube map projections, autoregressive strategies, and spherical operators struggle to achieve high-fidelity generation because seams and poles remain problematic. To address these issues, the authors introduce tangent-plane factorization for the pretrained SD3.5 to reduce distortion, consistency-aware refinement and grid optimization to remove seams, and distortion-aware metrics (TangentFID and TangentIS). The authors apply tangent-plane factorization from SphereDiff to fine-tune the DiT weights of SD3. Consistency-aware refinement includes ERP-conditioned latent denoising, circular padding (from Diffusion360 and PanoDiff), and a patchifying method for high-resolution generation. Grid optimization enhances the attention flow by calculating the cost of grid arrangement. During inference, a pretrained SR model (VARSR) is optionally used for high-resolution generation. The authors claim that the proposed model achieves SOTA performance in panoramic image generation, both qualitatively and quantitatively.

**Strengths:**

1. By leveraging prior approaches, TanDiT achieves SOTA performance in panoramic image generation.
2. The proposed metrics, TangentFID and TangentIS, appear to be promising indicators for evaluating distortion-aware panoramic image quality.
3. Extensive experiments are conducted to validate the effectiveness of each proposed component.

**Weaknesses:**

1. Limited novelty. Most components appear to be adapted from prior work, and the overall contribution seems largely engineering-oriented rather than conceptually new.
2. The qualitative results do not show clear visual improvements over existing baselines. Please include zoom-in views to highlight the claimed advantages.
3. What would be the performance of TanDiT if trained solely on the Matterport3D dataset? Since PanFusion and UniPano were trained only on Matterport3D, the inclusion of PolyHaven and Flickr360 introduces a fairness concern that should be clarified.
4. Please specify which dataset or image set was used as the reference for computing evaluation metrics.
5. The comparison currently includes fewer SOTA methods than expected. Please incorporate citations to recent SOTA models [1, 2, 3, 4, 5, 6]. At minimum, a quantitative and qualitative comparison with SMGD [5] and SaFa [6] should be added for completeness.
6. Please provide qualitative examples corresponding to high and low TangentFID/TangentIS scores. While the user study suggests quantitative alignment with human preferences, readers cannot visually interpret how these metrics reflect perceptual quality.
7. In Line 249, if the ERP latent is partitioned into 1024×1024 patches, does this correspond to a decoded resolution of 8192×8192? Please confirm or clarify.
8. The term “Latent rotation” first appears in L446 without prior definition. Is this referring to the “grid optimization” step? Please ensure consistent terminology with the method section.
9. Similarly, the term “Patch denoising” first appears in L449. Please ensure consistent terminology with the method section.

[1] Tang et al., MVDiffusion: Enabling Holistic Multi-view Image Generation with Correspondence-Aware Diffusion, NeurIPS 2023

[2] Lee et al., SyncDiffusion: Coherent Montage via Synchronized Joint Diffusions, NeurIPS 2023

[3] Quattrini et al., Merging and Splitting Diffusion Paths for Semantically Coherent Panoramas, ECCV 2024

[4] Lee et al., SemanticDraw: Towards Real-Time Interactive Content Creation from Image Diffusion Models, CVPR 2025

[5] Sun et al., Spherical Manifold Guided Diffusion Model for Panoramic Image Generation. CVPR 2025

[6] Dai et al., Latent Swap Joint Diffusion for 2D Long-Form Latent Generation, ICCV 2025

**Questions:**

Please see the weaknesses.

**Details Of Ethics Concerns:**

No concern.

---

### Official Review · Reviewer_HdsY · 2025-11-05

**Soundness:** 2
**Presentation:** 2
**Contribution:** 2
**Rating:** 4
**Confidence:** 5

**Summary:**

This paper proposes a method for synthesizing panoramic scenes by generating multiple grids of tangent-plane images and then stitching them together to create a seamless panoramic view. Post-processing steps are specifically designed to improve the quality of synthesized results, including an optional super-resolution module and re-noising operations. New distortion-aware metrics, TangentFID and TangentIS, are also proposed.

**Strengths:**

1. The writing is clear and easy to follow.

2. Splitting panoramas into multiple tangent plane grid patches to reduce projection distortion seems reasonable.

**Weaknesses:**

1. I am confused about the motivation and the difference between cube-map projections and the proposed tangent plane design. It seems that TanDiT degenerates into a structure similar to the standard cubemap when the number of tangent planes equals 6.

2. This paper mentions several relevant works in Sec.1 and Sec.2. However, only a part of them are compared in Tab.2. I am curious why the authors do not compare TanDiT with related works, including CubeDiff[1], PanoLlama[2], PAR[3], and Omni2[4]. In L57, the author claims that AR-based methods suffer from slow inference. The authors are encouraged to conduct inference time comparisons with these methods, including PanoLlama[2] and PAR[3].

3. In L154, the author claims that tangent patches approximate perspective images with minimal distortion for small field-of-view. However, splitting a whole panorama into too many patches harms global coherence. This concern is exacerbated at high resolutions, as the authors propose breaking down high-resolution images into multiple sub-images in L249.

4. The refinement module seems tricky and brings unfair gains compared to other diffusion-based methods. Also, the heavy reliance on post-processing methods hinders the implementation of end-to-end models.

5. In L310, the authors introduced a hyperparameter of 1.96 for calculating TangentIS and TangentFID, and I would like to know how this was calculated. Furthermore, it seems that performing cubemap projection and then calculating the FID separately could overcome the claimed seam and pole problem, and a similar method has already been implemented in MVDiffusion[5].

6. Typos. PanoLlama in L56 and PanoLLaMA in L107 are inconsistent.

[1] CubeDiff: Repurposing Diffusion-Based Image Models for Panorama Generation.

[2] PanoLlama: Generating Endless and Coherent Panoramas with Next-Token-Prediction LLMs.

[3] Conditional Panoramic Image Generation via Masked Autoregressive Modeling.

[4] Omni2: Unifying Omnidirectional Image Generation and Editing in An Omni Model.

[5] MVDiffusion: Enabling Holistic Multi-view Image Generation with Correspondence-Aware Diffusion.

**Questions:**

1. Can you explain why you chose SD3 rather than Flux, and train the model on a mixed dataset rather than a single dataset?

---

### Author Response · Authors · 2025-11-14

Despite most of the reviewers’ concerns being directly addressable or already answered in our paper, we are choosing to withdraw this paper, given the issues with reviews we address below.

We thank the reviewers for their time and efforts. We would like to point out that many of the referenced works mentioned by the reviewers are in fact not relevant to our work. The models PanoLLaMA, SyncDiffusion, SemanticDraw, MAD, and SaFa are all wide-image perspective panorama generation methods, and do not actually generate spherical/omnidirectional images like our approach. The key difference is that the ERP versions of omnidirectional images include polar distortions caused by projecting an image on a sphere into a rectangle. Furthermore, there is the requirement that spherical images be loop-consistent (moving off of the right side of the image should seamlessly transition to the left side of the image). MVDiffusion, on the other hand, does not generate polar regions, but instead aims to generate the equatorial regions of an ERP image, which renders it not directly comparable to our work. The training process of SMGD that takes weeks for a 2 staged training which trains a VQGAN and a generator conditioned on it caused us to drop it as a baseline in order to not present an unfair comparison, as the model weights shared by the authors typically did not give reasonable results when tested in our datasets. Finally, other works like MVDream and MVAdapter are not applied on panoramic images, and all the results shown in these papers are on vastly different applications and types of data, making it a challenging comparison, and not relevant to our work.

In addition, multiple of the mentioned works like SphereDiff and CubeDiff did not have code available at the submission deadline (SphereDiff released their code on October 8th, and CubeDiff only has a non-official implementation which does not necessarily guarantee identical results or fair comparisons with the original paper, and even this non-official implementation was only released on September 28th). Expecting comparisons with models having no publicly available code at the submission deadline is unreasonable. Additionally, CubeDiff arguably solves a much easier problem, image-to-omnidirectional image, since the reference image is able to provide much more detail about the desired scene than a simple textual description.

Finally, many of the other concerns mentioned by reviewers are already addressed in our paper or the supplementary. For example:

Reviewer ypY3 states “What would be the performance of TanDiT if trained solely on the Matterport3D dataset? Since PanFusion and UniPano were trained only on Matterport3D, the inclusion of PolyHaven and Flickr360 introduces a fairness concern that should be clarified.”. However, we explicitly stated in our paper “For fairness, we retrain all of the tested baselines on our dataset using their public implementations and recommended settings, ensuring that performance differences are attributable to model design rather than data mismatch.”, in Section 5 of the paper.

Reviewer HdsY states “I am confused about the motivation and the difference between cube-map projections and the proposed tangent plane design. It seems that TanDiT degenerates into a structure similar to the standard cubemap when the number of tangent planes equals 6.”. However, Section A.3.2 in the supplementary includes a detailed discussion about the differences between the cubemap and 18 tangent plane representations (including Table 4, which explicitly compares distortion levels between the two methods, showing that higher tangent planes like we use introduces less distortion when mapping between perspective images and parts of the sphere).

Reviewer ypY3 states “Please provide qualitative examples corresponding to high and low TangentFID/TangentIS scores. While the user study suggests quantitative alignment with human preferences, readers cannot visually interpret how these metrics reflect perceptual quality.” However, Figure 7 in the supplementary shows an example of multiple samples generated from a model getting high OmniFID but lower TangentFID/TangentIS scores. Since FID-based metrics are distribution-based and cannot be calculated on individual images, asking for specific qualitative examples is not possible.

Reviewer 4tuo asks “Are there cases where the refinement stage introduces new artifacts, such as duplicated objects or loss of semantic detail?”. However, Section H in the supplementary is explicitly titled Limitations, and includes a figure (Figure 20), which shows a specific example of where the refinement stage introduces duplicated objects.

---

### Note · Authors · 2025-11-14

**Comment:**

Please see the Official Comment below for our reasons for withdrawing.

**Withdrawal Confirmation:**

I have read and agree with the venue's withdrawal policy on behalf of myself and my co-authors.